# Neuronal cell cycle reentry events in the aging brain are more prevalent in neurodegeneration and lead to cellular senescence

Deng Wu[1], Jacquelyne Ka-Li Sun[1], Kim Hei-Man Chow [1,2,3] *

1 School of Life Sciences, Faculty of Science, The Chinese University of Hong Kong, Hong Kong SAR, China, 2 Gerald Choa Neuroscience Institute, The Chinese University of Hong Kong, Hong Kong SAR, China, 3 Nexus of Rare Neurodegenerative Diseases, The Chinese University of Hong Kong, Hong Kong SAR, China

* heimanchow@cuhk.edu.hk

**Data Availability Statement:** All original codes and metadata for each individual Figure are available in: https://zenodo.org/doi/10.5281/zenodo.10604562.

## Abstract

Increasing evidence indicates that terminally differentiated neurons in the brain may recommit to a cell cycle-like process during neuronal aging and under disease conditions. Because of the rare existence and random localization of these cells in the brain, their molecular profiles and disease-specific heterogeneities remain unclear. Through a bioinformatics approach that allows integrated analyses of multiple single-nucleus transcriptome datasets from human brain samples, these rare cell populations were identified and selected for further characterization. Our analyses indicated that these cell cycle-related events occur predominantly in excitatory neurons and that cellular senescence is likely their immediate terminal fate. Quantitatively, the number of cell cycle re-engaging and senescent neurons decreased during the normal brain aging process, but in the context of late-onset Alzheimer's disease (AD), these cells accumulate instead. Transcriptomic profiling of these cells suggested that disease-specific differences were predominantly tied to the early stage of the senescence process, revealing that these cells presented more proinflammatory, metabolically deregulated, and pathology-associated signatures in disease-affected brains. Similarly, these general features of cell cycle re-engaging neurons were also observed in a subpopulation of dopaminergic neurons identified in the Parkinson's disease (PD)-Lewy body dementia (LBD) model. An extended analysis conducted in a mouse model of brain aging further validated the ability of this bioinformatics approach to determine the robust relationship between the cell cycle and senescence processes in neurons in this cross-species setting.

## Background

Terminally differentiated neurons are believed to have irreversibly exited from the cell division process [1], and by various means, these cells obtain their postmitotic identity [2,3].

**Funding:** The work was supported, in part, by grants from the following: The Hong Kong Research Grants Council (RGC)-General Research Fund (GRF) (PI: ECS24107121, GRF16100219 and GRF16100718) (all to K.H-M.C) and the RGC-Collaborative Research Fund (CRF) (Co-I: C4033-19EF) (K.H-M.C); the National NaturalScience Foundation-Excellent Young Scientists Fund 2020 (Ref: 32022087) (K.H-M.C); Alzheimer's Association Research Fellowship (PI: AARF-17-531566) (K.H-M.C). All the funders had no role in study design, data collection and analysis, decision to publish, or preparation of the manuscript.

**Competing interests:** The authors have declared that no competing interests exist.

**Abbreviations:** AD, Alzheimer's disease; CDK, cyclin-dependent kinase; CNA, copy number alteration; CNV, copy number variation; DDR, DNA damage response; ES, early senescent; EST, expressed sequence tag; HMM, hidden Markov model; ND, nondemented; PD, Parkinson's disease; LBD, Lewy-body dementia; LS, late senescence; OPC, oligodendrocyte precursor cell; PC, principal component; PCNA, proliferating cell nuclear antigen; RADC, Rush Alzheimer's Disease Centre; rPCA, reciprocal principal component analysis; ROS, Religious Order Study; ROSMAP, Rush Memory and Aging Project; snRNA-seq, single-nucleus RNA sequencing; t-SNE, t-distributed stochastic neighbor embedding.

Nonetheless, a large body of work has indicated that as individuals age, a small population of these cells may recommit to a cell cycle-like process [4], and increasing levels of these events are found in disease-affected brains of patients with late-onset Alzheimer's disease (AD) [4–11]. While the reprogression and completion of the entire process of mitosis has never been reported, evidence related to the reexpression of cell cycle proteins [6–11] and somatic copy number variations (CNVs) [12] have been found [13–15], suggesting that a partial duplication of genomic materials might have co-occurred even though this process is likely prematurely halted by the activation of cell cycle checkpoint signaling through the physiological surveillance response [16]. While some studies have suggested that the rapid reexpression of cell cycle-related genes in neurons immediately leads to cell death and apoptosis [17,18], other reports have indicated that this phenomenon is instead a possible driver of the cellular senescence response [19–21]. To our knowledge, it remains unclear whether these consequences are mutually exclusive or instead a sequela of one another. Due to the randomness of the emerging locations and the limited quantities of these cells in the brain, their full characterization and molecular profiling via traditional histology-based or bulk tissue transcriptomic sequencing techniques have been challenging. To date, it remains unclear how these cell cycle re-engaging neurons may remain viable in the brain for months or even years [22,23] and whether they exhibit any disease-specific properties resulting from the varying brain microenvironments.

By taking advantage of the wealth of human brain single-nucleus RNA sequencing (snRNA-seq) datasets available in public repositories, we developed an analytical pipeline that facilitates the identification and characterization of cell cycle gene reexpressing neurons (Fig 1A) to address these questions. For this analysis, the cell cycle gene expression status of each single nucleus was identified to determine whether any postmitotic cell clusters exhibited enhanced expression of these genes. The related findings were subsequently validated by the InferCNV algorithm, as potential DNA duplication or deletion events might be due to aberrant cell cycle activity [24]. Subsequently, the identities of the target cells of interest were further characterized via cell fate trajectory analysis to uncover their origins and evolutionary relationships. To evaluate the possible roles of these genes in disease pathogenesis and phenotypic heterogeneity, we performed quantitative analysis of their relative numbers, and qualitatively, differential gene expression analysis was conducted for nuclei from control versus disease-affected samples.

## Results

### Identification of cell cycle re-engaging neurons in human brain snRNA-seq datasets

The human adult brain is composed of terminally differentiated cells, such as excitatory and inhibitory neurons [25], myelinated oligodendrocytes [26], and mature astrocytes [27], as well as other cells that readily exhibit proliferative propensities [e.g., oligodendrocyte precursor cells (OPCs), microglia, and endothelial cells]. To investigate whether the postmitotic status of these terminally differentiated cells was faithfully attained in aged healthy brains, we analyzed the snRNA-seq data of a discovery cohort [28] of 24 age- and sex-matched samples (Brodmann area 10 of the prefrontal cortex) harvested from individuals without dementia (ND) (S1 Table). Among a total of 34,140 nuclei (Fig 1B) in which all the cell types mentioned were identified based on various established markers (Fig 1C), 32,685 nuclei belonging to terminally differentiated cell types were selected and divided into 20 smaller subclusters (Fig 1D). Among these nuclei, those that had robustly re-expressed cell cycle genes were identified using the AddModuleScore function of Seurat, as each nucleus was assigned a "cell cycle phase score" based on the expression levels of a refined list of 357 cell cycle-related genes (S2 Table) collated

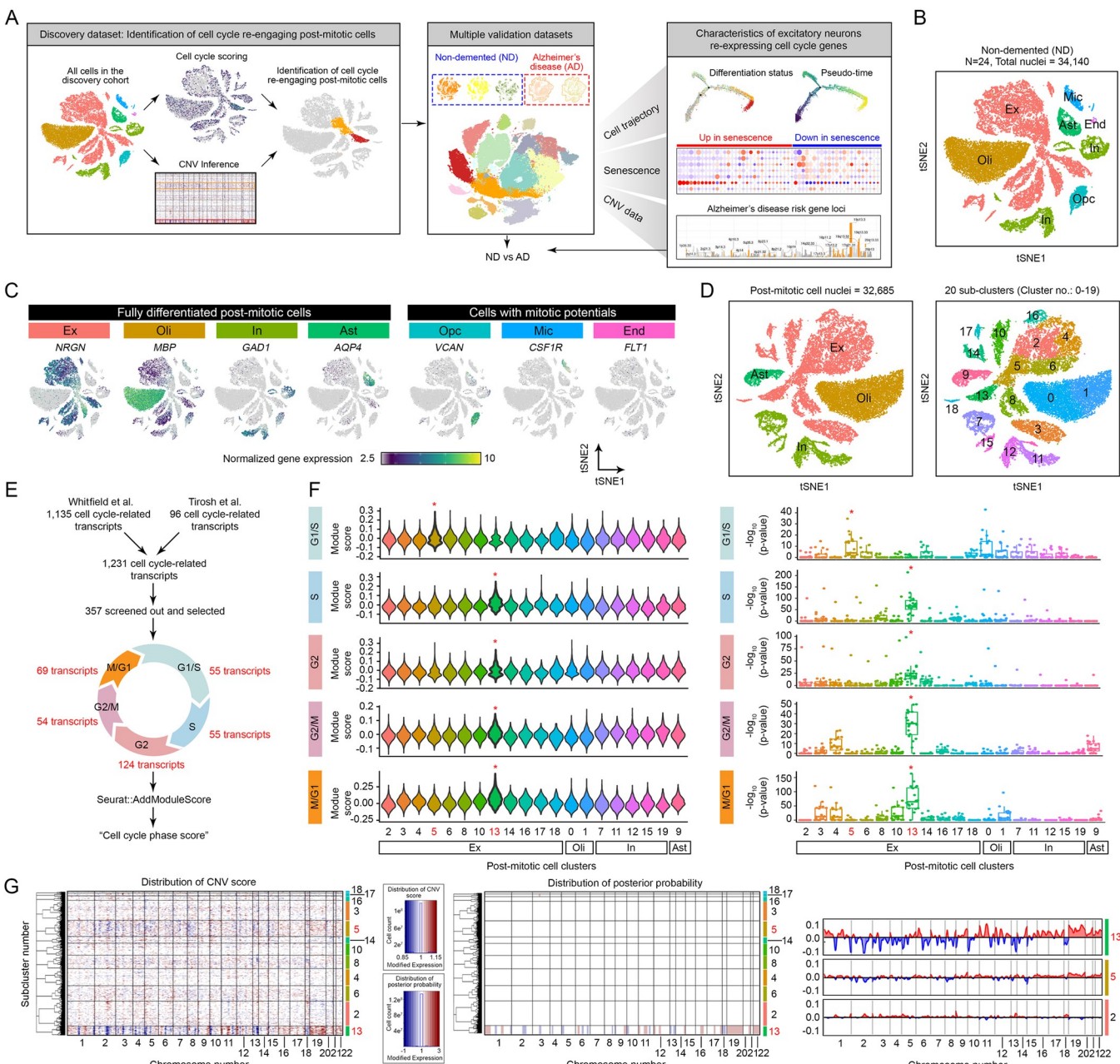

**Fig 1. Identification of cell cycle re-engaging postmitotic excitatory neurons in single-nucleus transcriptomic profiles.** (**A**) Schematic diagram illustrating the general layout and design of the bioinformatics analyses applied in this study. (**B**) t-SNE plot of a total of 34,140 nuclei derived from 24 unaffected, age (mean ± SD = 85.59 ± 4.20), and sex-matched (12 males and 12 females) prefrontal cortex samples from Brodmann area 10 [28]. The different cell types used are abbreviated as follows: excitatory neurons (Ex), oligodendrocytes (Oli), inhibitory neurons (In), astrocytes (Ast), microglia (Mic), endothelial cells (En), and oligodendrocyte progenitor cells (Opc). (**C**) Normalized expression levels of cell type-specific markers in all cell types mentioned above. (**D**) t-SNE plot of 32,685 postmitotic cell nuclei colored according to cell type (left panel) and subcluster number (right panel). (**E**) Schematic illustration of the cell cycle phase scoring workflow for each nucleus. (**F**) Violin plots showing the distribution of the cell cycle phase scores in all postmitotic cell subclusters. The bold highlights indicate the subcluster with the most significant differences above the average gene expression levels in any particular cell cycle phase. *P* values against other subclusters are shown on the right. (**G**) Estimation of copy number variants among all excitatory neurons via the InferCNV algorithm. The heatmap on the left shows the CNA regions identified by the HMM, i.e., regions of gain (red) and loss (blue) of expression along each chromosome, at various regions from the p-arm (left side of each box) to the q-arm (right side of each box) in all subclusters. The middle heatmap is an outcome of the Bayesian latent-mixture model implemented to identify the posterior probabilities of alteration status in each cell and whole CNA region. True positive predictions of CNV events were identified only in subcluster 13 (red: gain of copy number; blue: loss of copy number). On the far right are line plots illustrating in detail how gene expression levels on each chromosome are altered in subclusters 13 and 5 compared to a negative control (subcluster 2). The metadata underlying this figure can be found at https://zenodo.org/doi/10.5281/zenodo.10604562. CNA, copy number alteration; CNV, copy number variation; HMM, hidden Markov model; t-SNE, t-distributed stochastic neighbor embedding.

from the Whitfield and colleagues [29] and Tirosh and colleagues [30] studies (S2 Table), after which any hypothetical proteins, expressed sequence tags (ESTs) or alias names corresponding to the official gene symbols were screened out. After the corresponding cell cycle phase scores were assigned to each nucleus of all postmitotic cells, significant inductions in cell cycle genes were observed only among subclusters of excitatory neuronal nuclei [e.g., subclusters 5 (G1/S phase) and 13 (S, G2, G2/M and M/G1 phases)] (Fig 1F). For other postmitotic cell types, such as inhibitory neurons, myelinated oligodendrocytes, and mature astrocytes, this phenomenon was not found (Fig 1F).

Since the cell cycle is an ordered sequence of events [16], the enrichment of genes belonging to and downstream of the S phase in cell nuclei from only subcluster 13 suggested that these cells might have at least partially duplicated their DNA materials [31]. To predict whether such genome-level changes exist by studying the single-cell transcriptomic data, we used the inferring copy number variations (InferCNV) algorithm [32]. Among all the excitatory neuronal cell clusters, subcluster 13 was the only subcluster in which the InferCNV heatmaps predicted segmented CNVs (Fig 1G, $t$ test, $P < 2.2e-16$). In particular, significant gains in CNV scores were readily apparent across chromosomes 19 and 22 (Fig 1G, middle panel). Consistent with the cell cycle analysis, similar CNV signals were not detected among the other postmitotic cell types (S1 Fig). Similarly, among cells that retained proliferative capacities in the adult brain (e.g., OPCs, microglia and endothelial cells) (S2A–S2C Fig), CNV-related events were not detected (Fig 2D–2G), as any duplicated genomic materials in these mitotic cells is believed to be distributed evenly to their daughter cells by completing successful rounds of cell division. Together, these negative data further validated that cell cycle re-engagement events are unique to a certain subpopulation of excitatory neurons in the adult brain.

To validate the robustness of these observations, we integrated 3 validation cohorts of cerebral cortex samples (Brodmann areas 8, 9–12) from both individuals without dementia and those diagnosed with AD [28,33–35] into our existing analyses (Fig 2A and S1 Table). After correction for all batch effects, a total of 123,212 excitatory neuronal nuclei were extracted from 94 samples and were subsequently assigned to 15 subclusters (Figs 2B, S3A and S3B). The integrity of each subcluster was subsequently confirmed by the clustree function (S3A and S3B Fig), which confirmed that each subcluster was affected by neither dimension reduction (S3C–S3E Fig) nor integration methods (S3F–S3H Fig). In all subclusters, the relative number of nuclei in each individual dataset was calculated to confirm the unbiasedness of the batch distributions (Figs 2C, S4A and S4B), as well as the sex distribution of the samples (S4C Fig). In the subsequent analyses, subclusters 3 and 5 were characterized by a general loss in gene transcription activity, both in terms of the total number of genes expressed (Fig 2D) and the average transcript level abundance of each expressed gene (i.e., gene counts; S4D Fig). In contrast, these 2 clusters, however, exhibited the most robust expression of cell cycle phase genes, as reflected by their average cell cycle scores (S4E and S4F Fig). Among the 2 subclusters, subcluster 5 identified in this multi-dataset analysis was the only excitatory neuronal cluster reexpressing S-phase genes and genes downstream of this cell cycle phase (S4E and S4F Fig). Consistent with our earlier predictions from the discovery cohort (Fig 1G), InferCNV analysis predicted significant signs of CNV (e.g., CNV status >1.50 or <0.50) in this cluster (Fig 2E and 2F) compared to the negative control subcluster 0 located adjacent to subclusters 3 and 5 (which shared the highest similarities) on the tSNE plot (Fig 2B). Overall, these observations were alternatively validated when the validation cohorts were separately analyzed (S5 Fig). To predict how such changes may manifest and affect the physiological properties of excitatory neurons (e.g., subcluster 5), specific gene loci located within the predicted CNV gain (1,798 genes) and loss (911 genes) regions were identified (S3 Table). Pathway enrichment analysis indicated that genes located within the predicted CNV gain regions were primarily related to

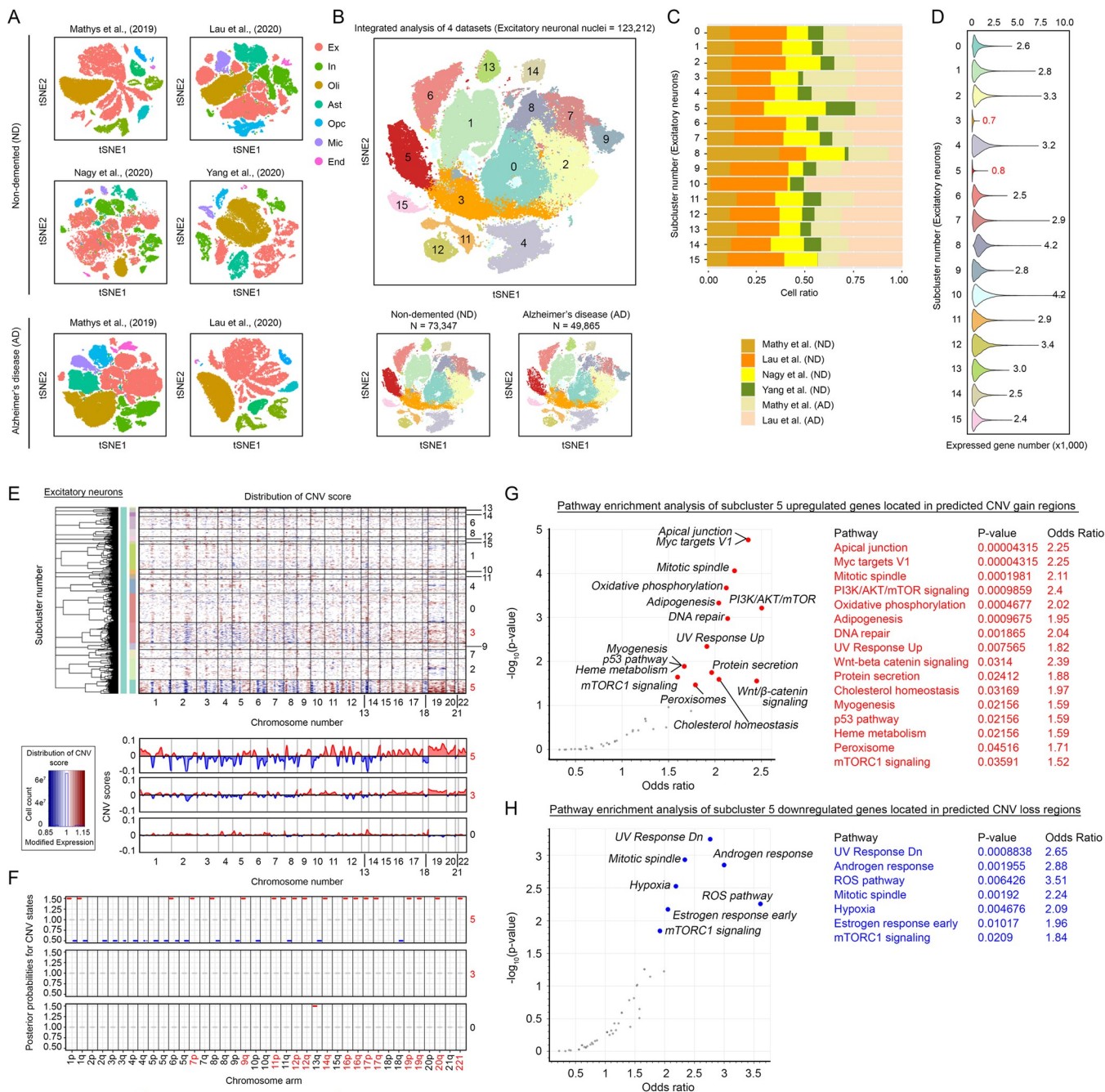

**Fig 2. Validation analyses with multiple cohorts of datasets and characterization of genes located in regions with predicted CNV events. (A)** t-SNE plots of all nuclei from a total of 4 datasets containing cohorts of ND and AD brain samples. The coloring code denotes different cell types. **(B)** t-SNE plot of 123,212 excitatory neuronal nuclei extracted from both ND and AD brain samples from the 4 independent datasets. **(C)** A proportional stacked bar graph illustrates a random distribution of cells from the 4 datasets in all the subclusters defined. **(D)** Violin plot indicating the total number of expressed genes detected among all 15 subclusters of excitatory neuronal nuclei. **(E)** Estimation of copy number variants among all excitatory neuronal nuclei extracted from the 4 datasets by the InferCNV algorithm. The heatmap on the top is an outcome of the Bayesian latent mixture model implemented to identify the posterior probabilities of alteration status in each cell and whole CNA region. True positives of CNV events were identified in subcluster 5 and, to a much lesser extent, in subcluster 3 (red: gain of copy number; blue: loss of copy number). At the bottom are the line plots that illustrate in detail how gene expression levels at different regions of each chromosome are altered in subclusters 5 and 3 in comparison to a negative control (subcluster 0). **(F)** The absolute CNV scores within each cytogenetic band were aggregated and are presented. A CNV state value of 1.00 was regarded as zero gain or loss. A change in value of 0.5 indicates a significant gain (e.g., 1.50) or loss (e.g., 0.50) of copy number in each chromosome arm. Plots belonging to subclusters 5, 3, and 0 are shown. **(G–H)** The genes located at the predicted CNV gain or loss regions of subcluster 5 were extracted (S3 Table) and were functionally clustered. The top scatter plot **(G)** illustrates pathways enriched from genes located in the predicted CNV gain regions, whereas the plot at the bottom **(H)** illustrates those enriched from the predicted CNV loss

regions. The metadata underlying this figure can be found at https://zenodo.org/doi/10.5281/zenodo.10604562. AD, Alzheimer's disease; CNA, copy number alteration; CNV, copy number variation; ND, nondemented; t-SNE, t-distributed stochastic neighbor embedding.

mitogenic signaling (e.g., PI3K/AKT/mTOR signaling, Wnt/β-catenin signaling, and mTORC1 signaling); cell cycle events (e.g., Myc targets V1 and mitotic spindle); the DNA damage response (e.g., DNA repair, UV response up, and p53 pathway); and metabolic reprogramming (e.g., oxidative phosphorylation, adipogenesis, and cholesterol homeostasis) (Fig 2G), while those identified from CNV loss regions were related to the sex hormone response (e.g., androgen response and early estrogen response) and reactive oxygen species signaling (Fig 2H). These data suggested that the molecular properties of cell cycle re-engaging excitatory neurons were substantially different from those of other cells.

## Extensive loss of neuronal function is associated with a gain of senescence related to erroneous cell cycle gene reexpression

The above observations repeatedly suggested that excitatory neurons, as the only postmitotic brain cell type, are detectable and viable after re-engagement of any erroneous cell cycle-related events. To determine the origins of these cells, we analyzed the expression levels of layer-specific markers corresponding to cerebral cortex cytoarchitecture [36]. A spatial transcriptomic sequencing matching analysis (S6 Fig) confirmed the layer-specific properties of all excitatory neuronal subclusters (Fig 3A–3C). Notably, these analyses suggested that subcluster 3, which re-expressed the early G1/S phase marker (S4E and S4F Fig), indeed best matched the upper II-III layer and II-IV layer neurons [e.g., **II-III**—*GLRA3 and TLE1;* **II-IV**—*CUX1, CUX2 and KITLG*] (Fig 3A). In contrast, the same set of analyses indicated that subcluster 5, which robustly re-expressed late cell cycle phase markers (S4E and S4F Fig), was more similar to a distinct group of neurons, as they no longer exhibited any obvious layer-specific identities (Figs 3A–3C and S6). Given that cell cycle progression and differentiation are inversely related processes [37], these observations also suggested that these 2 groups of neurons exhibit a gradual decrease in neuronal differentiation. To validate this hypothesis, we performed single-cell trajectory and branching analyses [38], and the branching locations of both subclusters of interest were shown to be molecularly distinct from the rest of the excitatory neuronal populations, as also validated separately in all the datasets (Figs 3D and S7). Notably, the neighboring localization of 2 subclusters on the same pseudotime scale branch (Figs 3D and S7) also suggested that they were evolutionarily related. For further analysis, pathways enriched from the overrepresented cell cycle genes found in these 2 clusters were compared, revealing that pathways involving DNA damage checkpoints [e.g., E2F targets [39], G2/M checkpoint [40], mitotic spindle [41], UV response, and DNA repair] were commonly activated (Fig 3E and 3F and S4 Table). Intriguingly, detailed evaluation of the status of the DNA repair response pathway revealed that a substantial number of genes encoding core elements of the mismatch repair, homologous recombination, and Fanconi anemia repair pathways were uniquely down-regulated in subcluster 5 compared with subcluster 3 and the remaining non-cell cycling excitatory neuronal nuclei (S8A–S8C Fig). These differences suggested a reduction in the number of pathways associated with DNA repair in these cells. Similarly, bipartite gene expression patterns were also observed among the key elements of other repair pathways, such as nonhomologous end joining, nucleotide excision repair, base excision repair, ubiquitin-related pathways, and poly(ADP-ribose) polymerase (PARP) enzyme regulation (S8D–S8I Fig). These observations also suggested that DNA lesions addressed by these partially inhibited pathways are at risk of repair infidelity.

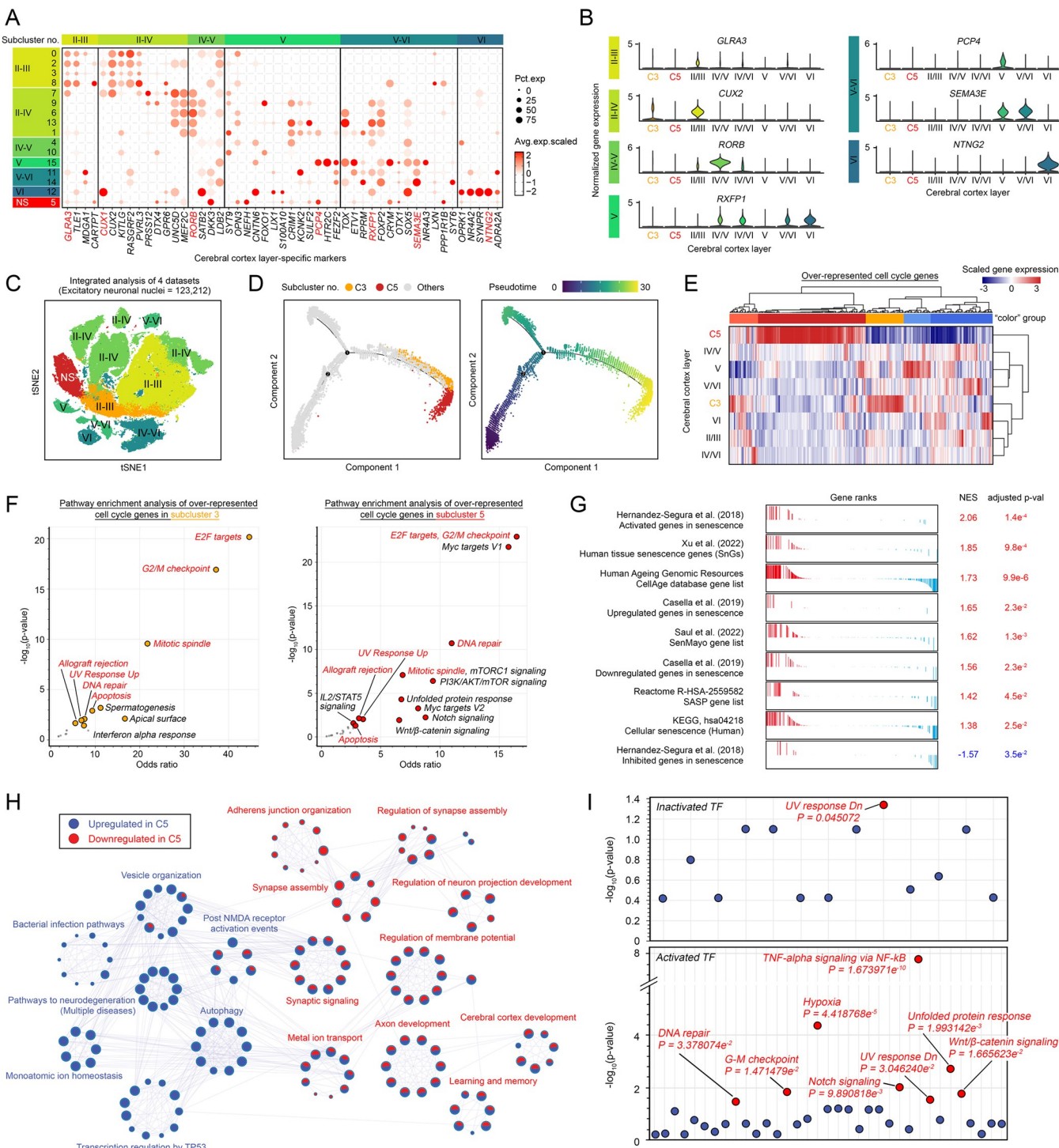

**Fig 3. Senescence is the immediate terminal fate of cell cycle gene reexpressing neurons. (A)** Dot plot showing the expression levels of cerebral cortex layer-specific markers among all excitatory neuron clusters. **(B)** Violin plots illustrating the expression levels of selected cerebral cortex layer-specific marker genes (highlighted in red in Fig 3A) in different subclusters. **(C)** t-SNE plot of excitatory neurons colored according to their cortical layer identity. NS = nonspecific. **(D)** Single-cell trajectory analysis with the Monocle 2.0 algorithm was used to determine the evolutionary relationship between the 2 subclusters of cell cycle gene-expressing neurons. The locations of subclusters 3 and 5 on this trajectory are labeled, with subcluster 5 located at the terminal. Similar findings were observed when each dataset was separately analyzed, as presented in S7 Fig. **(E)** Heatmap showing the relative expression levels of 357 cell cycle-related genes among all the excitatory neuron clusters. Hierarchical clustering of these genes based on their expression levels resulted in 5 colored groups. Red- and yellow-colored gene clusters represent up-regulated cell cycle gene groups in clusters 5 and 3, respectively. **(F)** Functional pathway enrichment analyses of cell cycle gene clusters (red and yellow) with reference to the Molecular Signatures Database (MSigDB) on the Enrich R platform [42,43]. **(G)** Gene set enrichment

analysis of multiple gene sets related to senescence signatures and phenotypes in subcluster 5. **(H)** Metascape enrichment analysis of all the DEGs identified in subcluster 5 compared to the other excitatory neuronal populations. The functional pathways enriched in the up- and down-regulated DEGs are illustrated in blue and red, respectively. **(I)** TFs that regulate DEGs that do not match the inferred CNV regions in subcluster 5 were predicted based on the SCENIC analysis, as presented in S11 Fig (S7 Table). Manhattan plots illustrating the enriched signaling networks of coinhibited (top panel) or coactivated (bottom panel) TFs. Significantly ($p < 0.05$) enriched networks are labeled. The metadata underlying this figure can be found at https://zenodo.org/doi/10.5281/zenodo. 10604562. TF, transcription factor; t-SNE, t-distributed stochastic neighbor embedding.

Indeed, DNA repair infidelity, which manifests as persistent activation of the DNA damage response (DDR) due to unresolvable DNA lesions (e.g., DNA repair infidelity), is a strong trigger of cellular senescence—a permanent state of cell cycle arrest [20,44] and possibly the cell fate of subcluster 5. In addition, the expression of 7 independently curated gene sets [45–49] implicated in cellular senescence was found to be significantly enriched in this subcluster (Figs 3G and S9A and S5 Table), and this phenomenon was also likely associated with the activation of the E2F1-RB axis, which could subsequently induce the reexpression of multiple cyclins and cyclin-dependent kinases (CDKs) (Figs 3F, S9B and S9C). Other possible senescence response mechanisms, including the compromised ATM/ATR signaling axis that coordinates the fidelity of the downstream DDR (S9D Fig) and the induction of senescence mediator CDK inhibitors [e.g., *CDKN1A* (p21), *CDKN1B* (p27<sup>Kip1</sup>), and *CDKN2C* (p18)], were also observed in these cells (S9E Fig). Phenotypically, these cells were also likely to acquire a collagenolytic phenotype [50] characterized by down-regulated expression of collagen components (e.g., *COL19A1*, *COL25A1*, *COL28A1*, *COL11A2*, *COL14A1*, *COL4A3*, *COL6A6*, *COL4A6*, *COL4A4*, and *COL6A5*), an activated secretory profile of mitogens (growth factors) (e.g., *VEGFA*, *FGF7*), and regulatory factors (e.g., *IGFBP2*, *IGFBP6*, and *IGFBP3*) that may promote hypermitogenic arrest [51] and the acquisition of prosurvival/antiapoptotic phenotypes [52] (e.g., *HSP90AB1*, *BCL2L1*, and *FOXO4*) (S9F Fig). Taken together, these findings suggested that the excitatory neuronal nuclei in subcluster 5 likely committed to a full senescence response; therefore, we defined these cells as "late senescent (LS)" neurons in the following analyses. Similarly, subcluster 3, located adjacent to these LS neurons on the tSNE plot (Fig 3C) and along the same trajectory branch (Figs 3D and S7), was considered their precursor and was therefore renamed "early senescent (ES)" neurons.

Phenotypically, in an attempt to further determine the salient and unique features of fully senescent neurons compared to nonsenescent neurons, differential gene expression analysis was performed (S6 Table), and post hoc system-level pathway network analysis via the Metascape algorithm [53] was performed. As predicted, the majority of the down-regulated DEGs in LS neurons were involved in axon development, the regulation of membrane potential and synaptic functions that together participate in cerebral cortex development and function (e.g., learning and memory) (Fig 3H, red labeling). The up-regulated DEGs were involved in pathways that lead to neurodegeneration (multiple diseases), stress responses (e.g., the bacterial infection pathway, monoatomic ion homeostasis, and transcriptional regulation by TP53), and autophagic events (e.g., autophagy and vesicle organization) (Fig 3H, blue labeling). Moreover, 33.6% of the up-regulated DEGs were indeed common to the list of genes located within the predicted CNV gain regions, with key functions in cell cycle events, such as mitotic centrosome and tubulin modeling, occurring during the G2/M phase (S10A and S10B Fig and S6 Table). Similarly, 28.8% of the down-regulated DEGs detected also matched the list of genes located within the predicted CNV loss regions, with roles in neuronal differentiation and function, axonal guidance and synaptic signaling (S10C and S10D Fig and S6 Table). These observations therefore suggested that the phenotypic changes associated with LS neurons were at least in part associated with selective chromatin remodeling and DNA replication events occurring at specific chromosomal regions. For the remaining DEGs that were not located in

the predicted CNV gain or loss regions, changes could emerge from altered activities of various transcription factors in these cells. As revealed by the SCENIC algorithm [54], a total of 162 activated transcription regulons were identified to regulate these genes (S7 Table). A total of 115 coactivated and 47 coinhibited transcriptional regulons were found (S11A Fig and S7 Table). However, although meaningful functions could not be identified from the coinhibitory set of transcriptional regulons, except for *PPARG* and *NR3C1*, which are related to the "down-regulated response to ultraviolet radiation" (S7 Table), transcription factors among the coacti-vated regulons play important roles in TNF-alpha signaling via NF-B [55], the hypoxic response [56], the unfolded protein response [57], the G2-M checkpoint [58], Wnt/β-catenin signaling [21], and the DNA repair network [59], which has been previously reported to be associated with the cellular senescence response (Fig 3I). These findings were faithfully repeated when each individual dataset included was separately analyzed (S11 Fig), confirming the robustness of these observations.

## Identification of disease-associated signatures among senescent neurons that emerged from AD brains

Aging is considered one of the most prominent risk factors for late-onset AD, suggesting that aged and senescent neurons resulting from aberrant cell cycle-related events may also contrib-ute to disease-related pathogenesis. To dissect how these cells are related to AD risk, we per-formed mapping analysis of their CNV profiles against an array of known disease risk gene loci [60] and an investigation of their relative expression levels. Since CNV is linearly associ-ated with gene expression levels for the majority of genes [61], their gain or loss of change can be indirectly predicted from gene expression analysis with reference to controls. With both the integrated (Fig 4A) and separated (S12 Fig) analyses, elevated expression levels of multiple AD risk genes were identified within the predicted CNV gain regions in LS neurons. This phenom-enon was in stark contrast to what was observed in neurons that did not re-express any cell cycle genes [e.g., subcluster 0 (sharing the highest similarities)] located adjacent to LS and ES neuron on the t-distributed stochastic neighbor embedding (t-SNE) plot (Fig 2B). With refer-ence to the matching trend of changes observed in previously reported AD-related CNV loci [12], including those found to be gained (e.g., *SDF4*, *BIN1*, *ERMP1*, and *VDLR*) (S13A Fig) and lost in AD (e.g., *SLC30A3*, *DNAJC5G*, *TRIM54*, *KLK6*, *HAS*, and *FPR1*) (S13B and S13C Fig), which validates our approach, an additional set of differentially expressed AD risk genes was identified. Examples of these genes include multiple major histocompatibility complex molecules, such as *HLA-A*, *HLA-B*, *HLA-C*, *HLA-DQB*, and *HLA-DRB1* (6p21.3), which may introduce selective vulnerability to neurons [62]; *NYAP1* (7q22.1), which modulates tau pathology [63]; and *CLU* (8q21.1), which promotes amyloidogenesis [64] (Fig 4B); and those that may directly contribute to the production of amyloid and related peptide fragments [e.g., *APP* (21q21.3), *APLP1* (19q13.12), *APLP2* (11q24.3), and *BACE1* (11q23.3)], which are all induced among LS neurons (Fig 4B). Additionally, the expression of both *MAPT* (17q21.31) and *APOE* (19q13.32) was induced (Fig 4B), which may lead to mitochondrial dysfunction [65] and the accumulation of amyloid-β at cortical synapses, respectively [66]. One exception was *PSEN1* (14q24.2), which encodes one of the 4 core proteins of γ-secretase; however, this gene was suppressed (Fig 4B). Together, these phenomena suggest that the emergence of cell cycle re-engaging neurons in the brain may contribute to AD-related pathogenesis. Quantita-tively, the relative number of these cells in the brain was under better control during the nor-mal aging process; however, in disease-affected subjects with AD, these cells tended to accumulate instead (Fig 4C). Given that tissue accumulation in senescent cells is generally pre-vented by the actions of local immune cells triggered by the detection and interactions of cell

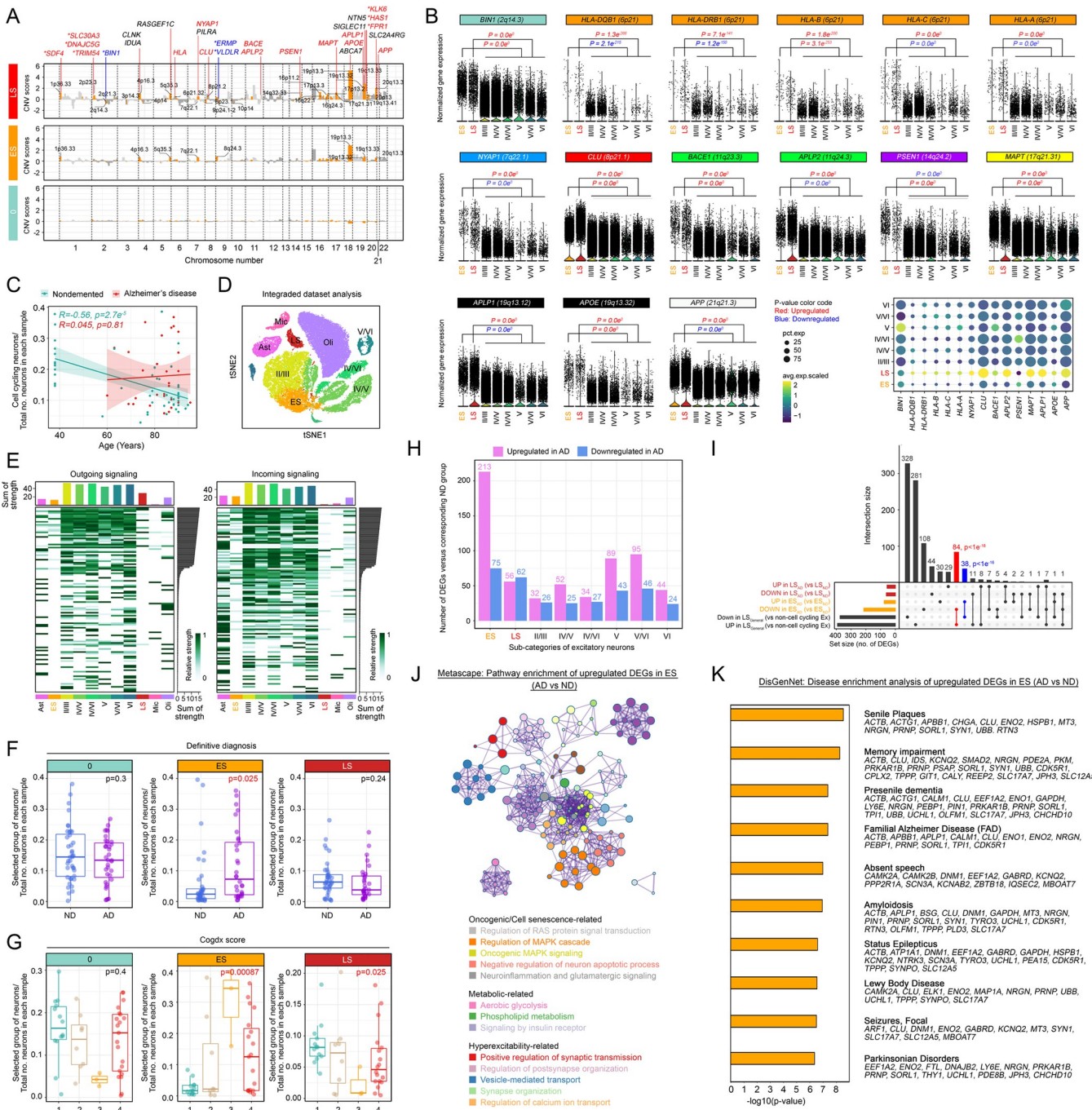

**Fig 4. Early senescent neurons in the AD brain exhibit a unique proneuropathological signature. (A)** AD risk gene loci mapped to CNV locations identified in subclusters 3 (ES) and 5 (LS) compared to the negative control subcluster 0. The yellow highlights indicate chromosomal locations of the classic AD risk gene loci where significant changes in the absolute CNV scores were observed compared with those of the control cluster. The red and blue labels indicate the up- and down-regulated AD risk genes, respectively, located within the highlighted locations. **(B)** Normalized expression levels of highlighted AD risk loci indicated in **(A)** among ES, LS, and the remaining non-cell cycle re-engaging neurons stratified in different cortical layers. **(C)** Scatter plot illustrating the correlation between the cell cycle re-engaging neuron ratio and the total number of neurons in each sample with age. **(D)** UMAP plot of excitatory neuronal and glial nuclei extracted from the Mathys and colleagues dataset [28]. The color code represents different cell types and neurons located at different cortical layers. **(E)** Heatmaps illustrating the differences in intercell communication strength among different cell types, quantified by corresponding ligand (left) or receptor (right) gene expression levels. The details of this analysis are shown in S1 Data files. **(F, G)** Dot and box plots illustrating the relationships between the cell number ratios of subclusters of concern and total neuronal nuclei in each sample with **(F)** a definitive diagnostic status and **(G)** Cogdx scores. **(H)** Bar plot representing the numbers of DEGs between groups of nuclei from AD versus ND samples in subclusters ES and LS and those belonging to different cortical layers. **(I)** Upset plot displaying the intersections among DEGs identified in the LS $_{(AD\ vs\ ND)}$, ES $_{(AD\ vs\ ND)}$, and LS $_{(general\ versus\ non-cell\ cycling\ neurons)}$ populations.

**(J)** Functional enrichment analysis of up-regulated DEGs in ES $_{(AD\ vs\ ND)}$ was performed on the Metascape platform. **(K)** Bar plot showing the disease enrichment analysis of up-regulated DEGs in ES$_{(AD\ vs\ ND)}$ with reference to the DisGenNet database. The metadata underlying this figure can be found at https://zenodo.org/doi/10.5281/zenodo.10604562. AD, Alzheimer's disease; CNV, copy number variation; ES, early senescent; LS, late senescence.

surface signals [67,68], further analysis of the expression levels of neuronal cell surface ligands and receptors involved in bidirectional communication with glia revealed that significant reductions in the expression of these transcripts were observed in cell cycle re-engaging neurons (e.g., ES and LS neurons) (Figs 4D, 4E and S14). This finding supported the notion of compromised recognition and subsequent elimination of these cells by brain-resident immune cells.

Our data suggest that cell cycle re-engaging and senescent neurons originating from healthy and affected brains share many commonalities. However, the accumulation of these cells in AD (Fig 4C) suggested that certain levels of heterogeneity may exist due to the extracellular influence of different brain microenvironments. However, while no significant associations were found between the overall cell number ratio and patient *APOE* status (S15 Fig) or neuropathological changes (e.g., Braak or CERAD) (S16 and S17 Figs), the specific cell number ratio of ES neurons was consistently correlated with definitive diagnostic status (e.g., AD, Figs 4F and S18) or severity of cognitive decline (e.g., higher Cogdx scores, Figs 4G and S19). These observations thus suggested that the phenotypic heterogeneities that could also exist were senescence stage specific. Further comparative analysis of the number of DEGs found in different subgroups of excitatory neuronal nuclei revealed that ES neurons from the AD brains (i.e., ES$_{AD}$) exhibited the most dramatic changes compared to those from ND brains (i.e., ES$_{ND}$) (Fig 4H). In AD samples, ES$_{AD}$ neurons exhibited general features of LS neurons (e.g., significant DEG overlap between ES$_{AD}$ and LS$_{General}$; Fig 4I), while LS$_{AD}$ neurons from the affected brains were also atypical (e.g., no significant overlap between LS$_{AD}$ and LS$_{General}$; Fig 4I). In ES$_{AD}$ cells, although the 75 down-regulated DEGs identified from the comparison with ES$_{ND}$ neurons failed to cluster into any meaningful signaling network, pathways enriched from the 213 up-regulated DEGs suggested that signaling networks related to oncogene-induced senescence (e.g., RAS and MAPK signaling, negative regulation of apoptotic process, and neuroinflammation), anabolic fuel metabolism (e.g., aerobic glycolysis, phospholipid metabolism, and insulin signaling), and hyperexcitability-related pathways (e.g., induction of synaptic transmission, postsynaptic organization, and calcium ion transport) were activated (Fig 4J). Further analysis of how this set of up-regulated DEGs may correlate with the entire landscape of human diseases [69] was conducted with reference to the DisGeNET database covered by the Metascape platform [53]. The results consistently highlighted the roles of these genes in multiple aspects of AD pathogenesis [e.g., the top 2 pathways: senile plaque formation (*ACTB*, *ACTG1*, *APBB1*, *CHGA*, *CLU*, *ENO2*, *HSPB1*, *MT3*, *NRGN*, *PRNP*, *SORL1*, *SYN1*, *UBB*, *RTN3*) and memory impairment (*ACTB*, *CLU*, *IDS*, *KCNQ2*, *SMAD2*, *NRGN*, *PDE2A*, *PKM*, *PRKAR1B*, *PRNP*, *PSAP*, *SORL1*, *SYN1*, *UBB*, *CDK5R1*, *CPLX2*, *TPPP*, *GIT1*, *CALY*, *REEP2*, *SLC17A7*, *JPH3*, *SLC12A5*)] (Fig 4K and S8 Table). Together, the activated pathological network in these cells (e.g., ES$_{AD}$) in part explained how their increased cell number ratio was correlated with more advanced functional impairment and disease pathogenesis (Fig 4K).

## Profiling of cell cycle re-engaging neurons in (1) human Parkinson's disease (PD)-Lewy-body dementia (LBD) and (2) mouse brain aging models

To validate the applicability of this analytical approach in other brain regions or disease settings, we also extended our analysis to the Parkinson's disease (PD)-Lewy-body dementia (LBD) model, as PD is the second most common age-related neurodegenerative disorder after

AD [70]. In PD-LBD, cell cycle-related events were previously reported in the dopaminergic (DA) neurons of the midbrain and dorsal striatum regions [71]. Based on this preexisting information, these neurons were first identified and selected [72] (S21A and S21B Table), followed by substratification into 9 subclusters (from 0–8) according to their transcriptomic signatures (S20A Fig). Subsequently, cell cycle scoring (S20B Fig), cell fate trajectory (S20C Fig), InferCNV profiling (S20D Fig), and targeted analyses of the expression levels of G2/M phase genes (S21C Fig) and evaluation of the expression status of 55 differentially regulated senescence genes defined by Hernandez-Segura and colleagues [47] (S20E Fig) were performed to identify subgroups of early and late senescent (LS) neurons. Consistently, these different methods suggested that subclusters 5 and 2 among all the dopaminergic nuclei exhibited features of ES and LS neurons, as defined earlier in Figs 1–4. To further address whether these ES and LS dopaminergic neurons that emerged from the affected (PD-LBD) mid-brains were molecularly distinct from those that emerged under unaffected conditions, DEG analysis was performed. Like in AD (Fig 4), the number of DEGs identified suggested that while the transcriptomic profiles of LS neurons (subcluster 2) that emerged from the PD-LBD and nonaffected groups were similar, a substantial number of differences indeed existed among the ES neurons (subcluster 5) that emerged from these different brain microenvironments (S20F Fig). Functional pathway enrichment analysis of this set of DEGs suggested that ES neurons in the PD-LBD brain microenvironment not only exhibited more extensive deregulation of dopaminergic synaptic functions but also revealed more obvious signs of mitochondrial hyperactivity [73,74] (e.g., oxidative phosphorylation and proton transmembrane transport), stress-induced senescence responses (e.g., oncogene-induced senescence and transcriptional regulation by TP53), and iron-related toxicity (e.g., iron update and transport) (S20G Fig). Alternatively, pathway enrichment analysis with reference to the DisGeNET also highlighted how this set of DEGs may contribute to protein aggregation [75] (e.g., amyloidosis), muscle and motor discoordination (e.g., Creutzfeldt–Jakob disease, sporadic; amyotrophic lateral sclerosis, muscle weakness, myopathy) [76] and dementia-related symptoms (e.g., familial AD; Parkinsonian disorders) that are commonly reported among PD-LBD patients (S20H Fig).

To further explore the application potential of this analytical approach in laboratory models, we performed additional analyses in a mouse model of brain aging [77] (S22A Fig). With a similar set of procedures, excitatory neuronal nuclei were again extracted and divided into smaller subclusters (S22B and S23A Figs). The cortical layer distributions were identified, except that of subcluster 2, as it barely expressed any layer-specific markers (S22C and S22D Fig). Subsequent cell cycle scoring (S22E Fig) and InferCNV analyses (S22F Fig) concordantly indicated that nuclei in subcluster 2 were cell cycle re-engaging with predicted signs of CNVs; moreover, this subcluster was uniquely characterized by a general loss in global gene transcription activities (S22G and S22H Fig). These multiple lines of evidence support that subcluster 2 identified from the aged mouse prefrontal cortex dataset closely resembles LS neurons, as defined previously in human dataset analyses (Figs 1–4). Subsequently, comparisons of both the up- and down-regulated gene signatures of these cells from mice and humans were conducted. To avoid double counting of any gene paralogs due to possible gene duplication [78], we first identified human and mouse gene orthologs prior to matching analysis (S22I Fig). A total of 372 up-regulated gene orthologs were identified from the original list of 392 human genes (S23B Fig and S9 Table), 162 of which (162/392 = 41.3%) were indeed common to the same marker gene list curated from the mouse dataset (S22I Fig) and implicated in enhanced mitochondrial oxidative phosphorylation and pyruvate metabolism (S22J Fig). These findings suggested that aberrant mitochondrial hyperactivity [73] and altered fates of pyruvate are conserved metabolic signatures of late senescent neurons. Similarly, of a total of 345 down-regulated gene orthologs identified from the original list of 375 human genes (S9 Table), 242 (242/

345 = 70.01%) were also common to those identified in mice (S22I Fig), with roles in regulating synaptic functions and axonogenesis (S22K Fig). The interrelationships between neuronal cell cycle gene reexpression, activation of the cellular senescence response, loss of mature neuronal identity, and an altered fuel metabolic (e.g., pyruvate metabolism) landscape were further validated by histology and immunohistology-based analyses. First, colocalization of the senescent cell marker SA-β-gal and the nuclear signal cyclin B [79] with the weakened signal of the postmitotic neuronal differentiation marker NeuN (e.g., *RBFOX1*) [80] first confirmed that the cells of interest were truly neurons (S22L and S24B Figs). Further evaluation of the key pyruvate metabolic pathway regulator pyruvate kinase (PKM) indicated that SA-β-gal+ cells were deficient in the constitutively active variant PKM1 in the cytosol but revealed positive signals of the PKM2 variant, which becomes metabolically defective when localized inside the nucleus (S22M, S24A and S24B Figs), suggesting that metabolic reprogramming might have occurred to facilitate the loss of mature neuronal identity among these cells [3,81]. Finally, SA-β-gal+ PKM1- cells were confirmed to be positive for the classic S-phase marker nuclear proliferating cell nuclear antigen (PCNA) (S22N and S24B Figs). Together with the global accumulation of the G2/M-phase marker Cyclin B [7] (S22L and S24B Figs), these findings also confirmed that the cellular senescence response is a concurrent event of cell cycle re-engagement in neurons.

## Discussion

Our study presents a readily applicable approach that allows the utilization of human or mouse brain single-nucleus transcriptomic data to decipher the molecular signature and cellular fate of cell cycle gene reexpressing neurons. The success in obtaining consistent findings in multiple independent datasets suggested that this analytical pipeline is a robust method that is applicable for systematically investigating the "cell cycling status" of all brain cell types by surveying their relative cell cycle gene expression levels and predicting possible CNVs. Our data indicated that the predicted signs of CNVs are consistent with a subset of excitatory neurons reexpressing S phase genes and those beyond this phase. Regardless of the disease status and brain regions of the samples, the predicted chromosomal locations of CNV events were not entirely random but were instead consistently concentrated in certain chromosomal regions (S5 Fig). We speculated that this difference is potentially related to the spatial organization of chromosomes inside human nuclei, which determines the ease of accessibility of DNA polymerases. For example, chromosome 19, which is frequently detected at the central position of the nucleus [82,83], was strongly associated with CNV gain events in cell cycle re-engaging neurons. In contrast, chromosome 18, which was detected more frequently at the periphery of the nucleus, was strongly associated with CNV loss events. Nevertheless, the lack of complete genome-wide duplication could at least in part be due to the concurrent initiation of S and G2/M DNA replication checkpoints in these cell cycle re-engaging neurons, which is a surveillance mechanism that prevents unchartered cell cycle activities in untransformed cells [84].

In addition to normal cell cycle arrest, cellular senescence is likely the terminal cell fate of cells reexpressing genes in the S phase and downstream genes, as this phenomenon was consistently observed in multiple dataset analyses. Regardless of the status of the samples (e.g., either unaffected or diseased-affected brains), these cell cycle re-engaging and senescent neuronal populations were molecularly distinct from their neighbors and characterized by a general loss of mature neuronal signatures and synaptic functions. Moreover, the expression of core genes involved in multiple DNA repair pathways was suppressed in fully senescent cells, which may lead to the repair infidelity that underlies the onset of the senescence response [19,20]. In the cerebral cortex, these cell cycle re-engaging and senescent neurons likely evolved from

pyramidal neurons located in layers II-III and III-IV. The latter is an important subtype of excitatory neurons because they interconnect brain association areas to shape numerous cortical functions; however, these cells are also selectively vulnerable to neurofibrillary tangle formation [85]. In support of these findings, a previous study indeed reported that tau protein aggregation is associated with neuronal senescence [86,87], and whether these events are linked to an unresolved DNA damage response and aberrant cell cycle events in humans warrants future investigation. In the context of late-onset AD, our data suggested that the relative quantities of early senescent neurons (e.g., those reexpressing only early G2/S cell cycle phase genes) were strongly associated with the definitive diagnosis of the disease and the severity of cognitive decline, which was in part due to the more extensive molecular heterogeneities that have existed in these cells that have emerged from the disease-affected brain microenvironment. In contrast to the cells found in nonaffected brains, those identified from diseased brains revealed many activated networks of proinflammatory and antiapoptotic pathways, metabolic reprogramming activities, and excitotoxicity-related responses. In addition, up-regulated genes found in early senescent neurons of affected brains were implicated in the formation of senile plaques, memory impairment, and other pathological features. These differences indeed suggested that the initial cell cycle re-engagement and subsequent deviations in molecular reprogramming may have led to heterogeneities in the properties that contributed to disease onset and progression.

In addition to its application in late-onset AD, our work also assessed whether this analytical pipeline is applicable for analyzing cell cycle re-engaging and senescent neurons in other human diseases (e.g., PD-LBD) and in cross-species settings (e.g., mouse models of brain aging). In the context of PD-LBD, cell cycle re-engaging and senescent dopaminergic neurons were identified in mid-brain tissues [88]. Likewise, the early but not the late senescent neurons that manifested in the PD-LBD brain microenvironment were substantially heterogeneous. These cells not only exhibited activated cellular senescence pathways but also hyperactivated mitochondrial respiratory activities [73,89], dysregulated iron homeostasis [90], and compromised dopaminergic synaptic activities that are uniquely implicated in parkinsonism [89]. Alternative disease pathway enrichment analysis also highlighted the potential involvement of these cells in protein aggregation [75] (e.g., amyloidosis), muscle and motor discoordination (e.g., Creutzfeldt–Jakob disease, sporadic; amyotrophic lateral sclerosis, muscle weakness, myopathy) [76], and dementia-related symptoms (e.g., familial AD; Parkinsonian disorders) that are commonly reported in PD-LBD. Similarly, in a mouse model of brain aging, this approach also identified these cells, which indeed exhibited features that were similar to those identified in the human brain, such as aberrant activation of mitochondrial respiratory activities and altered pyruvate metabolism, as well as decreased neuronal properties and synaptic functions. These similarities may be useful in explaining how advanced aging may serve as the greatest risk factor for AD and dementia [91].

Taken together, the results of this study illustrated a robust bioinformatics strategy that allows thorough investigation of the molecular profiles of cell cycle re-engaging neurons. The findings here demonstrated that cellular senescence is likely the immediate cell fate of these neurons, which also offered valuable insights into the molecular heterogeneities that could exist among these cells manifested in healthy and disease-affected brain microenvironments. While experimental validations of these findings in relevant human samples will be conducted in the future, the applicability of this analytical approach in different diseases and cross-species settings offers new opportunities and insights to supplement mainstay histological-based approaches in studying the roles of these cells in brain aging and disease pathogenesis. The data that illustrate disease-specific molecular signatures and the new marker genes detected in senescent neurons may also lead to new directions for future diagnosis and development of senotherapeutic strategies.

## Materials and methods

### Data sources and availability of data and codes

The following publicly available datasets were used: the Mathys and colleagues (syn3157322) [28] dataset downloaded from Synapse.org. The Nagy and colleagues (GSE144136) [34], Lau and colleagues (GSE157827) [33], Yang and colleagues [35] (GSE159812), Kamath and colleagues (GSE178265) [72], and Allen and colleagues (GSE207848) [77,92] datasets were downloaded from the GEO Omnibus. All analyses were carried out using freely available software packages. All the original codes for each individual figure are available at https://zenodo.org/doi/10.5281/zenodo.10604562.

### Cohort assembly from multiple single-nucleus transcriptome datasets

Both control (nondemented, ND) and affected (Alzheimer's disease, AD) samples were extracted from publicly available datasets from 4 independent studies: Mathys and colleagues [28], Nagy and colleagues [34], Lau and colleagues [33], and Yang and colleagues [35]. Only 2 datasets contained AD-related samples (Mathys and colleagues [28] and Lau and colleagues [33]).

According to the sample details extracted from the Mathys and colleagues study [28], all the samples belonged to participants in the Religious Order Study (ROS) or the Rush Memory and Aging Project (ROSMAP). The parameters they considered for categorizing samples as disease-affected (AD) and ND were primarily based on the degree of classic disease pathologies manifested in the samples. These included the "global AD pathology burden (gpath)" (average score: ND = 0.08 ± 0.08; AD = 1.51 ± 0.74), "overall amyloid level (amyloid)" (average score: ND = 0.03 ± 0.08; AD = 6.46 ± 2.31), and "neuritic plaque burden (plaq_n)" (average score: ND = 0.01 ± 0.04; AD = 1.79 ± 1.35). The definitions of these parameters are listed clearly on the Rush Alzheimer's Disease Centre (RADC) resource hub. The distinct numerical differences in the mean ± SD values in each of these domains indicated that the pathology-based classification of samples as AD versus ND was clear-cut without ambiguity.

On the other hand, according to the sample details extracted from the Lau and colleagues study [33], their ND and AD samples were also similarly defined; based on the distinct differences in disease-related pathologies, these included the "Age-related plaque scores" (e.g., ND = 0/A; AD = B/C; plaque load severity $_{\text{Low to High}}$ = A ➜ B ➜ C; 0 = nil) and "Braak tangle stage" (e.g., ND ≤ 2; AD ≥ 4).

As suggested by the values in both studies, the control ND patients were defined based on the absence of pathologies, whereas the AD patients were defined based on the obvious manifestations of heavy amyloid load and neuritic plaque burden. Details of the sample identities and demographics are listed in S1 Table. Since batch effects were detected among the datasets, batch effect removal was performed. These studies were also selected because all of them contained high proportions of excitatory neurons among all the nuclei sampled. These selection criteria increase the likelihood of identifying small subpopulations of cell cycle re-engaging neurons and increase the robustness of subsequent analyses. A PD/LBD dataset was later incorporated for cross-brain regional validation of the findings [72]. A mouse model of aging dataset was also included for cross-species application [77].

### General data processing, target neuron extraction and dataset integration

**1. General data processing.**   In each of the selected snRNA-seq studies, samples of different cell types were collected for clustering [28,33–35,72]. For identification of each of these genes, Seurat, an R package designed for the exploration of single-cell RNA-seq data for clustering different cell types, was used [93,94]. The filtered matrix of UMI counts and associated

row and column metadata for each dataset were downloaded from the corresponding links in the Synapse and GEO Omnibus databases. In the first step, a Seurat object was created with the count matrix, after which the genes were filtered based on their expression in at least 3 cells, and at least 200 genes had to be expressed in each cell for inclusion. All reads encoded from the mitochondrial genome were eliminated to avoid affecting the cell clustering results. After removing unwanted cells from the dataset, we log-normalized the entire dataset. Highly variable genes were identified, and a principal component analysis of the most variable genes was performed. An elbow plot was generated to select the principal components (PCs) capturing the most variance in the dataset. These PCs were used as edge weights in unsupervised, graph-based clustering to identify the top 30 cell clusters at a resolution of 0.5. t-SNE was used for visualization of all the cell clusters. The expression levels of cell type-specific markers were used to determine the putative identities of each cell cluster.

**2. Extraction of excitatory/dopaminergic neurons.** In both the Mathys and colleagues and Nagy and colleagues datasets [28,34], preexisting cell type assignments from their corresponding metadata were used for the extraction of excitatory neuronal nuclei in the samples, which was further confirmed by assessing the relative *NGRN* expression. For the Lau and colleagues and Yang and colleagues datasets [33,35], since the downloaded data did not include cell type assignments, annotations were assigned to the cells based on the expression levels of cell type-specific markers: neurogranin (*NRGN*) for excitatory neurons, glutamate decarboxylase-1 (*GAD1*) for GABAergic neurons, myelin basic protein (*MBP*) for myelinating oligodendrocytes, aquaporin 4 (*AQP4*) for astrocytes, versican (*VCAN*) for OPCs, colony stimulating factor 1 receptor (*CSF1R*) for microglia, and Fms-related receptor tyrosine kinase-1 (*FLT1*) for endothelial cells. Upon successful cell type labeling, excitatory neurons were extracted from the unaffected and affected samples and are presented as t-SNE plots. For the Kamath and colleagues dataset [72], dopaminergic neurons were extracted based on the cell type-specific markers vesicular monoamine transporter (*SLC18A2*) and tyrosine hydroxylase (*TH*).

**3. Dataset integration.** With all the excitatory neuron transcriptomic data extracted, these data were integrated into one by the Seurat integration method [94]. Reciprocal principal component analysis (rPCA) was used to identify anchors between any 2 datasets by using data extracted from the Mathys and colleagues study (with the largest number of samples) as the reference. The top 50 significant PCs derived from the elbow plot were used to locate anchors for integrating the datasets, while the top 30 PCs were selected to identify the neighbors. The resolution was selected for precise recognition of cell cycling clusters; therefore, a wide range of resolution values was tested. Clustering trees obtained by the clustree package [95] were used to facilitate the validation of cell clusters formed based on a range of resolutions. Starting from the low end, we first used a resolution of 0.5 for clustering potential cell cycle gene reexpressing neurons. As the value increased, further refinement of the clusters was allowed. Eventually, the highest resolution at 1.0 was used to validate the properties in the form of small clusters. t-SNE and UMAP plots used for visualizing all the integrated excitatory neuronal clusters. The dataset-wide distribution of sex, disease status, and cluster cell number distribution are shown.

On the other hand, the Harmony algorithm [96] was used to examine the influence of the integration method on excitatory neuron subcluster identification. The batch effects between datasets were corrected by the RunHarmony function in the Seurat package, with the dataset used as the group variable. The same parameters used in the rPCA integrated pipeline were adopted, including the top 30 PCs, and a resolution of 0.5 was selected to cluster the excitatory neurons. The consistency of subcluster constitutions between Harmony and the rPCA was evaluated by their shared nuclei and visualized using an alluvium plot.

## Cell cycle score analysis

Previously reported cell cycle-associated genes and their corresponding phase information were extracted from both the Whitfield and colleagues [29] and Tirosh and colleagues [30] studies. Within this combined list, only genes detected by single-nuclei sequencing were retained, while those annotating hypothetical proteins, ESTs, and gene aliases to official names were screened out. The core scoring function AddModuleScore in CellCycleScoring from the Seurat package was used to calculate a cell cycle phase-specific score for each cell. The permutation value set in the AddModuleScore was 100. The cell cycle phase score of each cell was collated and visualized as violin plots for identifying neuronal subclusters with higher expression of most cell cycle-associated genes, i.e., cell cycle re-engaging neurons. In each cell cycle phase, the cell cycle score of a subcluster was compared to that of the other subclusters to determine the significance of the differences. Cell cycle gene enrichment in different clusters, wherever applicable, was visualized via a heatmap. Clustered genes were then functionally annotated by the EnrichR platform, and the details are provided in the corresponding methods section. A schematic workflow of this analysis is illustrated in Fig 1E.

## InferCNV copy number detection and AD loci analysis

The InferCNV algorithm was used to predict copy number alterations (CNAs) from the snRNA-seq data (InferCNV of the Trinity CTAT Project: https://github.com/broadinstitute/InferCNV) [97]. To date, InferCNV is the only available method for exploring gene expression intensities across various chromosomal positions in the genome. A comparison of the gene expression intensities between cells of interest and a set of reference "normal" cells was performed to confirm the somatic changes observed. This method is therefore considered an ideal tool for identifying large-scale chromosomal CNAs in cell cycle re-engaging neurons via single-cell transcriptomic datasets. In brief, raw counts of 10,000 excitatory neurons were randomly sampled and selected from the corresponding Seurat object. The InferCNV algorithm was subsequently used to predict CNAs using the following settings: cutoff = 0.1 (this value was found to generally work well with 10X Genomics and 3′-end sequencing and droplet assays), HMM_type = 'i6′ (hidden Markov model (HMM)-based CNA prediction), and HMM_report_by = c("cell") (which instructs InferCNV to report per-cell level CNAs).

Given the CNA regions identified by the HMM, a Bayesian latent mixture model was implemented to identify the posterior probabilities of alteration status in each cell and whole CNA region. This method was used to combat possible missed identification by HHM of CNAs or cells that might not be true CNAs (false-positives). CNV regions identified by the HMM were filtered out if the posterior probability of the CNV region being normal exceeded a specified threshold. This step combines the possibility of misidentified CNVs by removing CNVs that were most likely to be normal and not true CNVs. By default, this threshold was set to 0.5, given that any CNV region that had a posterior probability of being in a normal state greater than 0.5 was relabeled "normal" and was no longer considered an identified CNV region. Changes to various chromosome arms were calculated by concatenating the CNV events detected on each arm according to the InferCNV pipeline and visualized by the point function in the ggplot2 package. The locations of the CNVs detected on the chromosome arms were obtained by using centromere and gene coordinate information from the GENCODE hg38, wherein the "end" was the first and last genes on each arm.

The cytogenetic band for each gene was downloaded from the HGNC database, and cytogenetic band-dependent CNV scores were defined as the sum of the CNV scores of the total genes located in the corresponding cytogenetic band. The results are presented as bar plots generated using the geom_bar function of the ggplot2 package. AD gene loci were manually

collected from a recent genome-wide association study [60], and the corresponding cyto-genetic bands are highlighted in yellow.

## Cortical layer identity evaluation

A list of cortical layer-specific markers was manually collated from the published literature [34,98]. The relationships among excitatory neuronal subclusters were first quantified by unsupervised clustering, followed by integrating the cortical layer markers across the subclusters. The comparative data are presented as layer-specific clusters, with 2 additional clusters indicating early- and late-senescent neurons. For validation of the true layer location of neurons annotated by different layer-specific markers curated from the literature, the expression levels of layer-specific markers across the cerebral cortical layers were quantified and visualized in spatial transcriptomics data [98] using the spatialLIBD package (v.1.13.4) [99]. Sample #151507 of the jhpce#HumanPilot10x dataset available at http://research.libd.org/globus [98] was used as the reference in our analysis. The vis_gene and layer_boxplot functions were used to visualize the layer-specific expression of different marker genes.

## Identification of subcluster-specific markers and DEGs between AD patients and ND patients

After the cortical layer identities were revealed, subcluster-specific markers were identified using the FindMarkers function of the Seurat package with the MAST method, which would have considered the dataset source, sex, and age as confounding factors. The markers of fully senescent neurons [i.e., late senescence (LS)] were obtained by comparing this cluster to the remaining excitatory neurons, considered only when the $p$ value $<0.05$ and the fold change $>$ |2|. For early senescent (ES) neurons, since their layer identity could be distinguished (early layer), marker genes were considered only when the $p$ value was $<0.01$ via comparison to the excitatory neuronal nuclei that also belong to the upper layers. On the other hand, DEGs between ADs and NDs in neurons in different cortical layers or senescent groups were identified using FindMarkers with the MAST method, which also considered the dataset source, sex, and age as confounding factors. Candidate gene markers with a $p$ value $<0.01$ were considered.

## Comprehensive gene set enrichment analysis with the EnrichR and Metascape platforms

EnrichR is an open-source intuitive enrichment analysis web-based tool that provides various types of visualization summaries of collective functions of genes [100] (http://amp.pharm.mssm.edu/Enrichr). This tool was used to identify gene-related biological processes, and signaling pathways with a threshold value of $p < 0.05$ and top combined scores were visualized and displayed.

Metascape is another comprehensive, biologist-oriented, publicly available resource (https://metascape.org/gp/index.html#/main/step1) for systems-level dataset analysis [53]. This tool was used to detect network-level changes and relationships among various significantly ($p < 0.05$) enriched pathways from more than 40 independent knowledge bases within the Metascape portal. Related analyses were visualized and displayed.

## Cell fate trajectory analysis with the Monocle 2.0 algorithm

Among the cerebral cortex samples, transcriptome profiles of excitatory neuronal nuclei extracted from all datasets were selected for analysis. Among the mid-brain samples,

transcriptome profiles of dopaminergic neuronal nuclei were extracted for analysis instead. For determination of the cell evolutionary trajectory among cell cycle gene reexpressing neurons, Monocle 2.0 [38] was used to order cells in a pseudotime manner based on the differentially expressed gene profiles. In brief, the genes differentially expressed between subclusters of excitatory neurons in each study were identified by the "differentialGeneTest," followed by DDRTree deconvolution, which returned a principal tree of centroids of cell clusters with reduced complexity in low dimensions.

## Cell–cell communication

CellChat, a tool that enables quantitative inference and analysis of intercellular communication networks from single-cell RNA-seq data, was applied to predict the major signaling inputs and outputs for cells based on the curated list of protein interactions, such as paracrine/autocrine ligands/receptors, extracellular matrix receptor interactions, and cell–cell contact interactions [101]. Relevant ligand (L) and receptor (R) genes were extracted from excitatory neurons, astrocytes, oligodendrocytes, and microglia. For every pair of cell types, the L-R interactions were identified and measured. The degree of intercellular communication was then predicted based on the projected L and R profiles, where the expression levels were approximated by their geometrical mean across individual cells of a type. These interactions represented the interaction strengths (also referred to as "probability" in CellChat) between all the ligands and their receptors expressed in 2 given cell types. Note that CellChat considers important signaling factors such as heteromeric complexes involved in each interaction in addition to an L-R pair; therefore, the absence of any of those components leads to a null interaction. Genes expressed in less than 20% of the cells in one cell type were excluded, and only statistically significant ($p < 0.05$, permutation test) communications were considered in our analysis. From individual L-R interactions to signaling, the interaction score of a signaling pathway was calculated by summing the interaction strengths for all the L-R interactions in the pathway. Dysregulated pathways among different neuronal layers were identified (Wilcoxon test, $p < 0.05$) and visualized in chord diagrams and dot plots.

## The regulon activity of transcription factors determined with the SCENIC algorithm

The single-cell regulatory network inference and clustering (SCENIC) algorithm was developed to conduct regulatory network analysis of transcription factors and discovery regulons (that is, transcription factors and their target genes) in individual cells [54]. In brief, a gene expression matrix was created with the gene names in rows and cells in columns as input for the python implementation of the SCENIC algorithm (pyscenic) [54]. A single pool of excitatory neurons merged from the 4 datasets included a total of 30,000 cells from random sampling of all the subclusters. The coregulation of transcription factors and their potential target genes was inferred using the grn function of the grnboost2 method. The ctx function was used to define the transcription factor-dependent regulons and to identify high-confidence target genes based on their *cis*-regulatory cues with the mask_dropouts parameter turned on. The AUCell command was subsequently used to calculate the regulon activity for each cell. Any transcription factors with AUC scores smaller than the 0.01 threshold among any cortical layers were omitted. The final AUC matrix was scaled across the cortical layers and was clustered using the ward. D2 method. The cisTarget database used for this analysis included the "hg38__refseq-r80__10kb_up_and_down_tss.feather" and "hg38__refseq-r80__500bp_up_and_100bp_down_tss.mc9nr.feather."

## Selected gene list analyses

Multiple lists of senescent genes were obtained from the Reactome database (R-HSA-2559583), KEGG database (HSA-04218), Xu and colleagues [49], Hernandez-Segura and colleagues [47], and Coppe and colleagues [102]. The human DNA damage response gene set was extracted from https://www.mdanderson.org/documents/Labs/Wood-Laboratory/human-dna-repair-genes.html [103] for DNA repair fidelity assessment. Layer-specific markers of the prefrontal cortex layers, as indicated by a list of cell surface markers implicated in immune surveillance [104], were ordered by a combination of their relative expression levels and percentage of gene-expressing cells in each subcluster. Dot plots were generated by the DotPlot function in the Seurat package, with normalization by the ggplot R package. A heatmap was generated with the pheatmap R package.

## Clinical trait association analyses

Clinical traits and disease status of the samples, whenever available, were extracted from the details provided along with the transcriptome datasets. The associations with the total cell number in different neuronal clusters were visualized as boxplots generated by the ggplot R package. Associations between age and the respective number of senescent neurons in the same sample were displayed by the point function in ggplot. Correlation coefficients were estimated by the Pearson method.

## Animal maintenance and brain tissue harvesting

C57BL/6J mice of both sexes were obtained from the Jackson Laboratory. Mouse colonies were maintained and bred at the Laboratory Animal Services Centre of CUHK. All animal experimental protocols were approved by the Animal Ethics Committees at CUHK (Approval number: 19/243/GRF), and their care was in accordance with both the institutional and Hong Kong guidelines. Brain tissue harvesting was performed by first anesthetizing the mice via the intraperitoneal administration of 1.25% (vol/vol) Avertin at 30 ml/kg body weight. The heart of each mouse was then surgically exposed, the left chamber was catheterized, and the right atrium was opened. Chilled physiological saline was perfused transcardially for 3 min to remove blood from the body. After perfusion, the cranial bones were opened; cortex and cerebellum tissues were harvested, snap-frozen in liquid nitrogen, and stored at −80˚C until use.

## Reagents for immunostaining

Unless otherwise specified, all chemicals and reagents were purchased from Sigma-Aldrich. All primary antibodies used were purchased from Thermo Fisher, and the sequences of the antibodies used were as follows: anti-PKM1 antibody (15821-1-AP), anti-PKM2 antibody (PA5-28700), anti-NeuN antibody (Cat #702022), anti-PCNA (13–3900), and anti-cyclin B1 antibody (MA5-13128). Alexa Fluor secondary antibodies were also purchased from Thermo Fisher.

## Senescence-associated β-galactosidase assay with immunostaining

Senescence β-galactosidase staining was performed as we previously reported (Chow and colleagues) with a colorimetric kit according to the manufacturer's protocol. In brief, frozen specimens 10 μm thick were briefly fixed in 1% formaldehyde for 1 min, and staining was subsequently performed using the same procedures as those used for the cultured cells. Caution was taken in determining the incubation time in brain sections, as subpopulations of neurons in the brain will sometimes develop false-positive signals if the incubation time extends

beyond 12 h or overnight. Immunostaining was performed after the SA-β-gal staining procedures. In brief, sections were washed twice in PBS after the blue color developed and then permeabilized with 0.5% Triton X-100 in PBS for 5 min at room temperature. After that, the samples were blocked with 0.5% BSA in PBS for 1 h and incubated with the primary antibody in 0.5% BSA for another 3 h at room temperature. Once this incubation step was completed, the samples were washed 3 times with PBS (10 min each) and incubated with secondary antibodies in 0.5% BSA for 1 h at room temperature. The washing step was then repeated, followed by nuclear staining with 1 μg ml–1 4,6-diamidino-2-phenylindole (DAPI) solution for 5 min. Samples were washed twice in PBS, mounted, and observed under a microscope.

## Histology-based data quantification procedures and statistical analysis

For each experiment, no statistical methods were used to predetermine sample sizes, but our sample sizes were similar to those reported in our laboratory's recent publications [20,21]. The data were tested and confirmed to be normally distributed. All samples were analyzed, and the data collected were blinded to the experimental conditions. All the experiments were performed on at least 3 independent occasions. Quantification of cellular morphology parameters was performed in a blinded manner. Analyses of the immunohistochemistry data were performed on Prism 8.0. Differences between groups were analyzed using two-tailed unpaired Student's $t$ test (for 2 groups) for all normally distributed data.

## Supporting information

**S1 Fig. Postmitotic cells other than excitatory neurons revealed limited signs of predicted CNV events.** Estimation of copy number variants by the InferCNV algorithm in **(A)** inhibitory neurons, **(B)** oligodendrocytes, and **(C)** astrocytes. The heatmap located at the top of each panel indicates the copy number alteration regions identified by the hidden Markov model, i.e., regions of gain (red) and loss (blue) in expression along each chromosome at various regions from the p-arm (left side of each box) to the q-arm (right side of each box), in all subclusters. The heatmap located at the bottom of each panel is an outcome of the Bayesian latent mixture model implemented to identify the posterior probabilities of alteration status in each cell and whole CNA region. Red: gain of copy number. Blue: loss of copy number. (JPG)

**S2 Fig. Brain cells with reserved mitotic capacities revealed limited signs of predicted CNV events.** **(A, B)** t-SNE plot of 1,455 mitotic cell nuclei. **(A)** Cell types are colored differently: microglia (Mic), endothelial cells (En), oligodendrocyte progenitor cells (Opc), pericytes (Per), and T cells (T). **(B)** Cells are colored based on subclustering numbers. **(C)** Violin plots of the cell cycle phase scores of all subclusters of mitotic nuclei. Bolded violins in different phases indicate the subcluster with the most significant above-average gene expression levels among all the clusters. Corresponding significance values against other clusters are shown below. **(D–G)** Estimation of copy number variants by the InferCNV algorithm in **(D)** Opc, **(E)** En/Per, **(F)** Mic, and **(G)** T cells. The heatmap located at the top of each panel indicates the copy number alteration regions identified by the hidden Markov model, e.g., regions of gain (red) and loss (blue) in expression along each chromosome at various regions from the p-arm (left side of each box) to the q-arm (right side of each box), in all subclusters. The heatmap located at the bottom of each panel is an outcome of the Bayesian latent mixture model implemented to identify the posterior probabilities of alteration status in each cell and whole CNA region. Red: gain of copy number. Blue: loss of copy number. The metadata underlying this figure can be

found at https://zenodo.org/doi/10.5281/zenodo.10604562.
(JPG)

**S3 Fig. The definition of excitatory neuronal subclusters was not affected by resolutions, reduction dimensions, or dataset integration methods. (A)** Clustering tree diagram illustrating how excitatory neuronal subclusters were defined at different resolution settings. Each row represents the number of subclusters identified at one particular resolution, and each column represents the number of subclusters and consistencies across different resolutions. As expected, higher numbers of subclusters were defined under higher resolution settings. The target subclusters (e.g., #3 and #5) had already emerged since the resolution was set at 0.4, and their distinct identities held when the settings were increased to 0.7. As the resolution increased to greater than 0.7, these 2 major subclusters started to split into smaller subclusters; nevertheless, they never cross-mixed with others derived from non-cell cycle re-engaging neurons and vice versa. **(B)** t-SNE plots of excitatory neuronal nuclei subclustering in the 4-dataset integrated analysis settings at different resolutions corresponding to those illustrated in **(A)**. **(C)** t-SNE and **(D)** UMAP plots of excitatory neuronal subclusters generated by the rPCA integration method at a resolution of 0.5. **(E)** UMAP plot of the excitatory neuronal subclusters generated by the t-SNE dimension reduction method. **(F)** t-SNE plot of the excitatory neuronal subclustering generated by the Harmony integration method at a resolution of 0.5. **(G)** Excitatory neuronal nuclei were colored and labeled according to the findings from the rPCA integration method, as illustrated in **(C)**. **(H)** Alluvial diagram showing the intersecting subclusters defined by the Harmony (left) versus the rPCA (right) integration methods. The resulting high consistency in subcluster identity indicated that excitatory neuronal cluster identification was not affected by the integration methods or other analytical settings. The metadata underlying this figure can be found at https://zenodo.org/doi/10.5281/zenodo.10604562.
(TIF)

**S4 Fig. Distribution of neuronal cells based on dataset source, sex, and disease status, as determined by cell cycle analysis in the integrated analysis setting. (A)** t-SNE plots of excitatory neuronal nuclei extracted from nondemented (ND) samples from different studies. **(B)** t-SNE plots of excitatory neuronal nuclei extracted from disease-affected (AD) samples from Mathys and colleagues and Lau and colleagues. **(C)** t-SNE plots of the excitatory neuronal nuclei distribution based on sex and disease status. **(D)** Violin plot illustrating the average feature counts of global transcriptomic profiles among excitatory neurons in different subclusters. **(E)** Violin plots presenting the cell cycle phase scores of all subclusters of excitatory neurons. Bolded violins highlighted in different phases indicate the subclusters that exhibit the most significant above-average cell cycle gene reexpression among all the subclusters. The corresponding significance values obtained for each subcluster compared to the rest of the others are shown in **(F)**. The metadata underlying this figure can be found at https://zenodo.org/doi/10.5281/zenodo.10604562.
(TIF)

**S5 Fig. Prediction and profiling of copy number variation separately in multiple datasets. (A, B)** Estimation of copy number variants in excitatory neuronal clusters extracted from either **(A)** nondemented (ND) or **(B)** disease-affected (AD) brain samples by the InferCNV algorithm. Red: gain of copy number. Blue: loss of copy number. **(C, D)** Line plots illustrating in detail how gene expression levels at different chromosome locations were altered in subclusters 3 and 5 compared to those in the negative control subcluster 0 extracted from either **(C)** nondemented (ND) or **(D)** disease-affected (AD) brain samples. The metadata underlying this

figure can be found at https://zenodo.org/doi/10.5281/zenodo.10604562.
(TIF)

**S6 Fig. Validation of cortical layer markers by spatial transcriptomic analysis. (A)** Visualization of cortical layers in sample #151707 from the jhpce#HumanPilot10x dataset using spatialLIBD. **(B)** Visualization of the distribution and counts of cortical layer-specific markers per spot. **(C)** Boxplots showing the expression levels of various layer-specific markers across different spatial locations defined in **(A)** to validate their layer specificities. **(D)** t-SNE and **(E)** UMAP plots illustrating the differential enrichment of cortical layer-specific markers in different neuronal subclusters in the integrated cohort analyses.
(TIF)

**S7 Fig. Single-cell trajectory analysis was performed using Monocle 2.0.** Cells on the trees are colored based on cell states, subcluster identities, cortical layer distributions, senescent neuronal cluster assignments, and pseudotime scales. Subclusters 3 and 5 were found along the same branch and were deemed to be the most similar to one another based on their pseudotime values. Subcluster 3, located at the terminal location of a branch, indicated terminal cell fate. The data represent findings from **(A)** nondemented (ND) or **(B)** disease-affected (AD) samples.
(TIF)

**S8 Fig. Compromised DNA damage response network in late senescent neurons. (A–I)** Normalized expression levels of core DNA damage response genes implicated in **(A)** mismatch repair, **(B)** homologous recombination, **(C)** Fanconi anemia, **(D)** nonhomologous end joining, **(E)** nucleotide excision repair, **(F)** base excision repair, **(G)** ubiquitination and modification, **(H)** poly(ADP-ribose) polymerase (PARP) enzymes that bind to DNA, and **(I)** other conserved DNA damage response genes. The metadata underlying this figure can be found at https://zenodo.org/doi/10.5281/zenodo.10604562.
(TIF)

**S9 Fig. Additional marker signatures of late senescent neurons. (A)** Expression levels of 55 differentially regulated senescence genes defined by the Hernandez-Segura and colleagues study [47]. **(B–F)** Expression levels of selected and classic genes involved in the **(B)** E2F-RB axis, **(C)** Cyclin-CDK axis, **(D)** ATM-Chk2/ATM-Chk1 axis, **(E)** CDK inhibitor, and **(F)** senescence-associated secretory phenotype profile obtained from the study by Coppe and colleagues [102]. The metadata underlying this figure can be found at https://zenodo.org/doi/10.5281/zenodo.10604562.
(TIF)

**S10 Fig. Impact of predicted CNVs on the transcriptomic signature in subcluster 5 (late senescent neurons). (A, C)** Venn diagram illustrating the degree of similarity of the DEGs identified in subcluster 5 compared to the remaining non-cell cycle gene reexpressing excitatory neurons in the list of genes located in the predicted **(A)** CNV gain and **(C)** loss regions. **(B, D)** Functional overrepresentation analysis of common genes identified in **(A)** and **(C)**, respectively, with reference to pathways in the Reactome database. The metadata underlying this figure can be found at https://zenodo.org/doi/10.5281/zenodo.10604562.
(TIF)

**S11 Fig. Characterization of upstream events underlying the transcriptome profile changes in terminally senescent neurons. (A)** In an integrated cohort setting, an SCENIC binary regulon activity matrix showing that all 162 corrected regulons were activated in more than one subcluster. Each column represents neurons in a single cortical layer (or ES or LS neuronal

clusters), and each row represents one regulon. The term "regulon" refers to the regulatory network of transcription factors and their target genes. Key regulons (rows) are magnified and colored according to their activities: active (orange) or inactive (blue) in the ES and LS neuronal clusters. **(B)** The same set of analyses illustrated in (A) was conducted separately for each individual dataset. Manhattan plots illustrating the enriched signaling networks of coinhibited (top panel) or coactivated (bottom panel) TFs. Significantly ($p < 0.05$) enriched networks are labeled. The metadata underlying this figure can be found at https://zenodo.org/doi/10.5281/zenodo.10604562.
(TIF)

**S12 Fig. AD risk gene loci mapping to predicted CNV locations in subclusters 5, 3, and 0.** The yellow highlights indicate chromosomal locations of classic AD risk gene loci. The metadata underlying this figure can be found at https://zenodo.org/doi/10.5281/zenodo.10604562.
(TIF)

**S13 Fig. Normalized expression levels of canonical AD-related CNV loci [12].** Comparisons were made among ES, LS, and the remaining non-cell cycle re-engaging neurons stratified based on their cortical layer identity. Reference list of CNV genes that were found to be **(A)** lost or **(B)** gained in AD. **(C)** Dot plot presenting the scale average expression levels of these canonical AD-related CNV genes among all the clusters compared. The metadata underlying this figure can be found at https://zenodo.org/doi/10.5281/zenodo.10604562.
(TIF)

**S14 Fig. Cell cycle gene reexpressing and senescent neurons exhibit compromised two-way communication with glia.** Dot plots illustrating the detailed results of the connectome analysis, which indicate the probabilities of communication between the indicated pairs of ligands and receptors according to the color label. The strength of the interaction, as indicated by the presence of a dot and the color between ligands expressed on glia and cell surface receptors expressed on neurons in various clusters, are shown on the left. Similarly, cell surface receptors expressed on glia and ligands expressed on neurons in various clusters are shown on the right. The metadata underlying this figure can be found at https://zenodo.org/doi/10.5281/zenodo.10604562.
(TIF)

**S15 Fig. Correlation plots between the cell number ratios of different neuronal subclusters and the sample *APOE* status.** The metadata underlying this figure can be found at https://zenodo.org/doi/10.5281/zenodo.10604562.
(JPG)

**S16 Fig. Correlation plots between the cell number ratios of different neuronal clusters and the sample Braak stage.** The metadata underlying this figure can be found at https://zenodo.org/doi/10.5281/zenodo.10604562.
(JPG)

**S17 Fig. Correlation plots between the cell number ratios of different neuronal clusters and the sample CERAD scores.** The metadata underlying this figure can be found at https://zenodo.org/doi/10.5281/zenodo.10604562.
(JPG)

**S18 Fig. Correlation plots between the cell number ratios of different neuronal clusters and the sample definitive disease diagnosis status.** The metadata underlying this figure can

be found at https://zenodo.org/doi/10.5281/zenodo.10604562.
(JPG)

**S19 Fig. Correlation plots between the cell number ratios of different neuronal clusters and the sample Cogdx scores.** The metadata underlying this figure can be found at https://zenodo.org/doi/10.5281/zenodo.10604562.
(JPG)

**S20 Fig. Application of the bioinformatics analytical pipeline in the Parkinson's disease (PD)/Lewy-body dementia (LBD) model. (A)** t-SNE plot of dopaminergic neurons extracted from healthy and diseased mid-brain samples; these nuclei were divided into 9 subclusters (0–8). **(B)** Violin plots showing the distribution of cell cycle phase scores in all dopaminergic neuronal nuclei subclusters. The bold highlights indicate the subcluster with the most significant above-average gene expression levels in any particular cell cycle phase; $p$ values against other subclusters are shown. **(C)** Single-cell trajectory analysis with the Monocle 2.0 algorithm revealing the evolutionary relationship between 2 subclusters of cell cycle gene-expressing neurons. Locations of subclusters 2 and 5 on this trajectory are labeled, indicating that they are on the same trajectory of fate, with subcluster 2 located at the terminal. **(D)** Estimation of copy number variants among all the dopaminergic neuronal nuclei extracted via the InferCNV algorithm. True positives of copy number variation events were identified in subcluster 2 (red: gain of copy number. Blue: loss of copy number). **(E)** Dot plot showing the expression levels of 55 differentially expressed senescence genes defined by Hernandez-Segura and colleagues [47] among all dopaminergic neuronal subclusters. **(F)** Bar plot representing the numbers of DEGs between groups of nuclei from PD-LBD patients and nondemented samples in all subclusters of dopaminergic neuronal nuclei. **(G)** Functional enrichment analysis of both up- and down-regulated DEGs in subcluster 5/ES (PD-LBD vs ND) was performed on the Metascape platform. **(H)** Bar plot showing the disease enrichment analysis of both up- and down-regulated DEGs in the ES cohort (PD-LBD vs ND) with the DisGenNet database. The metadata underlying this figure can be found at https://zenodo.org/doi/10.5281/zenodo.10604562.
(TIF)

**S21 Fig. Supporting data for the analysis performed with the PD/LBD dataset (S20 Fig). (A)** t-SNE plots of all nuclei derived from both the unaffected and affected samples in the Kamath dataset [105]. Clusters are colored according to cell type identity. **(B)** Dopaminergic neurons were identified and selected based on 2 classic markers: *SLC18A2* and *TH*. **(C)** Dot plot showing the scaled average expression levels of the classic G2/M genes among all the neuronal clusters. Elevated expression of most of these genes was identified in subclusters 5 (ES) and 2 (LS). The metadata underlying this figure can be found at https://zenodo.org/doi/10.5281/zenodo.10604562.
(JPG)

**S22 Fig. Application of the bioinformatics analytical pipeline in a mouse model of brain aging. (A)** UMAP plot of a total of 48,052 nuclei derived from frontal cortex samples harvested from four 21-month-old female C57BL/6J mice from the Allen and colleagues (GSE207848) dataset [77,92]. The different cell types used are abbreviated as follows: excitatory neurons (Ex), oligodendrocytes (Oli), inhibitory neurons (In), astrocytes (Ast), microglia (Mic), vascular cells (Vas), oligodendrocyte progenitor cells (Opc), microglia (Mic), and macrophages (Mac). **(B)** t-SNE plot of extracted excitatory neuronal nuclei divided into 16 subclusters (0–15). **(C)** Dot plot showing the expression levels of cerebral cortex layer-specific markers among all excitatory neuron clusters. **(D)** t-SNE plot of excitatory neurons colored according to their cortical layer identity. NS = nonspecific. **(E)** Violin plots presenting the cell cycle phase

scores of all subclusters of excitatory neurons. Bolded violins highlighted in different phases indicate the subclusters that exhibit the most significant above-average cell cycle gene reexpression among all the subclusters. *$p < 0.05$. **(F)** Estimation of copy number variants among all excitatory neurons via the InferCNV algorithm. The heatmap illustrates the copy number alteration regions identified by the hidden Markov model, e.g., regions of gain (red) and loss (blue) in expression along each chromosome, at various regions from the p-arm (left side of each box) to the q-arm (right side of each box) in all subclusters. **(G, H)** t-SNE plots indicating the **(G)** total number of expressed genes and **(H)** expressed gene counts detected among all 16 subclusters of excitatory neuronal nuclei. **(I)** UpSet plot displaying the intersections among marker genes identified in human and mouse LS neurons. **(J, K)** Pathway enrichment analyses of the **(J)** 242 common up- and **(K)** 162 down-regulated LS neuronal marker genes were conducted on the EnrichR platform. **(L)** Representative SA-β-gal and immunofluorescence staining images validating the interrelationship between cellular senescence, neuronal cell cycle reentry, and altered expression of PKM isozymes in the prefrontal cortex regions ($n = 10$; quantification is shown in S24B Fig; scale bar: 200 μm). The metadata underlying this figure can be found at https://zenodo.org/doi/10.5281/zenodo.10604562.
(TIF)

**S23 Fig. Supporting data for the analysis were generated with the mouse brain aging dataset (S22 Fig). (A)** Normalized expression levels of cell type-specific markers in all cell types mentioned in S22A Fig. **(B)** Dot plot showing the scaled average expression levels of common markers identified between human and mouse LS neurons. The metadata underlying this figure can be found at https://zenodo.org/doi/10.5281/zenodo.10604562.
(TIF)

**S24 Fig. Supporting data for the mouse brain histology analysis (S22L Fig). (A)** Representative SA-β-gal and immunofluorescence staining images validating the inverse relationship between PKM1 and PKM2 expression in neurons [$n = 10$; quantification is shown in **(B)**; scale bar = 200 μm]. **(B)** Quantification analyses of the brain histology data presented in S22L and S24A Figs ($n = 10$, ***$p < 0.0001$, unpaired $t$ test). The metadata underlying this figure can be found at https://zenodo.org/doi/10.5281/zenodo.10604562.
(TIF)

**S1 Table. Patient demographic details and related clinical information.**
(XLSX)

**S2 Table. Original cell cycle gene lists extracted from Whitfield and colleagues and Tirosh and colleagues studies and the final refined gene list.**
(XLSX)

**S3 Table. Lists of genes located at the predicted copy number alteration regions in subcluster 5 and pathway enrichment analysis.**
(XLSX)

**S4 Table. Overrepresented cell cycle genes in subclusters 3 and 5 and corresponding pathway enrichment analyses.**
(XLSX)

**S5 Table. Cellular senescence-related gene lists extracted from multiple databases and independent studies.**
(XLSX)

**S6 Table. Differentially expressed genes identified in subcluster 5 (late senescence) compared with the remaining excitatory neurons.**
(XLSX)

**S7 Table. Details of the transcription factor lists identified from the SCENIC algorithm via pathway enrichment analysis.**
(XLSX)

**S8 Table. Pathway enrichment analysis of up-regulated DEGs in ESAD with reference to DisGeNET.**
(XLSX)

**S9 Table. Common marker genes shared between human and mouse late senescent (LS) neurons and pathway enrichment analysis.**
(XLSX)

**S1 Data files. Supporting data for Figs** 1F, 1G; 2C–2H; 3A, 3E–3I; 4A–4C, 4F–4J; S2C; S3H; S4D and S4F; S5; S8; S9A–S9F; S10A–S10D; S11A, S11B; S12; S13; S14; S15; S16; S17; S18; S19; S20B, S20E–S20G; S21C; S22C and S22I; S23B; S24B.
(ZIP)

## Acknowledgments

The authors gratefully acknowledge the following parties. The results published here are in whole or in part based on data obtained from the AD Knowledge Portal related to the ROS-MAP metadata accessed at https://www.synapse.org/#!Synapse:syn3157322 and deposited by Mathys and colleagues. We would also like to acknowledge the GEO Omnibus for the datasets deposited by Nagy and colleagues (GSE144136), Lau and colleagues (GSE157827), Yang and colleagues (GSE159812), Karmath and colleagues (GSE178265), and Allen and colleagues (GSE207848). We would also like to thank Karl Herrup of the University of Pittsburgh and Ronald P. Hart of Rutgers University for helpful advice, support, and comments.

## Consent statement

The data available in the AD Knowledge Portal would not be possible without the participation of research volunteers and the contribution of the data by collaborating researchers. Informed consent was obtained from all individual subjects by the depositors of the data. An agreement on the data and sharing policies was signed to ensure proper protection of the privacy, application and publication of the deidentified data. No further consent or approval from the human data was needed.

## Author Contributions

**Conceptualization:** Deng Wu, Kim Hei-Man Chow.

**Data curation:** Deng Wu, Jacquelyne Ka-Li Sun, Kim Hei-Man Chow.

**Formal analysis:** Deng Wu, Kim Hei-Man Chow.

**Funding acquisition:** Kim Hei-Man Chow.

**Investigation:** Deng Wu, Kim Hei-Man Chow.

**Methodology:** Deng Wu, Kim Hei-Man Chow.

**Project administration:** Kim Hei-Man Chow.

**Resources:** Kim Hei-Man Chow.

**Software:** Kim Hei-Man Chow.

**Supervision:** Kim Hei-Man Chow.

**Validation:** Kim Hei-Man Chow.

**Visualization:** Kim Hei-Man Chow.

**Writing – original draft:** Deng Wu, Kim Hei-Man Chow.

**Writing – review & editing:** Kim Hei-Man Chow.

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
