## [Editor Report · Decision Letter 0]

22 Nov 2023

Dear Dr Chow, 

Thank you for submitting your revised manuscript entitled "Deciphering neuronal cell cycle re-entry and associated senescence events in healthy brain aging and dementia" for consideration as a Short Report by PLOS Biology.

Your manuscript has now been evaluated by the PLOS Biology editorial staff as well as by an academic editor with relevant expertise, and I am writing to let you know that we would like to send your submission back to the original reviewers. 

Once your full submission is complete, your paper will undergo a series of checks in preparation for peer review. After your manuscript has passed the checks it will be sent out for review. To provide the metadata for your submission, please Login to Editorial Manager (https://www.editorialmanager.com/pbiology) within two working days, i.e. by Nov 24 2023 11:59PM.

Kind regards,

Luke

Lucas Smith, Ph.D.

Senior Editor

PLOS Biology

lsmith@plos.org

---

## [Decision Letter · Decision Letter 1]

17 Jan 2024

Dear Kim,

Thank you again for your patience while we considered your revised manuscript "Deciphering neuronal cell cycle re-entry and associated senescence events in healthy brain aging and dementia" as a Short report at PLOS Biology. Your revised study has now been evaluated by the PLOS Biology editors, the Academic Editor and the original reviewers.

In light of the reviews, which you will find at the end of this email, we are pleased to offer you the opportunity to address the remaining points from the reviewers in a revision that we anticipate should not take you very long. As a note, we think that it would be important to add the requested analyses - and so we are happy to provide an extension as needed. We will then assess your revised manuscript and your response to the reviewers' comments with our Academic Editor aiming to avoid further rounds of peer-review, although might need to consult with the reviewers, depending on the nature of the revisions.

As you address the last reviewer comments, we also ask that you address the following editorial requests: 

1) TITLE: After some discussion within the team, we think the title could be strengthened if it conveyed more of the specific findings of the study. If you agree (and if supported), we suggest you change it to something like: 

"Neuronal cell cycle re-entry events in the aging brain are more prevalent in neurodegeneration and lead to cellular senescence"

2) ETHICS STATEMENT: Please update the ethics statement in your manuscript to provide the approval number for the protocol, approved by the ethics committee of CUHK. 

3) DATA/CODE: Thank you for providing the data and code underlying your manuscript on Github. Can you please generate a DOI for this dataset and code? This can be done with zenodo. https://docs.github.com/en/repositories/archiving-a-github-repository/referencing-and-citing-content

Can you please also add a brief sentence to each relevant figure legend referencing pointing readers to this repository. For example , you can add the sentence "the data underlying this figure can be found at ___" 

4) As a last note, we think that in places the manuscript could benefit from a quick edit for English (probably a bit beyond what copy-editing might catch). We suggest that you run the manuscript by a critical colleague to help improve this aspect as this will help make the data accessible to our broad audience. 

**IMPORTANT - SUBMITTING YOUR REVISION**

*Resubmission Checklist*

*Published Peer Review*

*PLOS Data Policy*

*Blot and Gel Data Policy*

Sincerely,

Lucas

Lucas Smith, Ph.D.

Senior Editor

PLOS Biology

lsmith@plos.org

REVIEWS:

Reviewer #1: My first comment, "It is unclear how the criteria for determining aged, non-diseased brains and AD brains were established. It is important for the criteria to be consistent across the four cohort studies to ensure accurate comparisons.", was not addressed in the revision though a supplemental table S1 was included. MY question is about how AD was determined in each of the four cohorts analyzed. If AD was defined differently, the analysis need be repeated based on the data with a consistent definition of AD cases.

Reviewer #2: The authors provided a comprehensive response to questions and suggestions. The inclusion of mouse data in the revision have further strengthened their interpretations. Only a few minor edits were noted and are listed below: 

* Abstract lines 25-30 is somewhat contradictory where the use of "substantial" is used to indicate that the process is common, but the following sentences describes these cells as "rare existences" and "rare cell populations." Remove the word "substantial" from line 25 as it is not needed, suggested edit, "Mounting evidence indicate that terminally differentiated neurons in the brain may re-commit to a cell cycle-like process under both normal aging and disease affected conditions."

* Line 83: Identification "of" cell cycle re-engaging neurons in human brain snRNA-seq datasets

* Line 130 is a bit confusing - needs editing for clarification.

* Line 151, suggest deleting the word "harvested"

* Line 208, replace "lost" with "loss"

* Line 210: neuron is should be "neurons as"

* Line 221: "Considered" should be "Considering"

* Line 527: To note, the study (Dehkordi et al 2021, Nature Aging; PMID: 35531351) also identified senescent excitatory neurons by evaluating the datasets evaluated here. Adding this reference would strengthens the conclusions presented here.

* Line 546: The study (Riessland et al 2018, Cell Stem Cell; PMID: 31543366) also found senescent dopaminergic neurons. Adding this reference will strengthen the interpretations from the PD dataset presented here 

Reviewer #3: I would like to emphasize the authors have dedicated effort to address most of the comments and critiques raised from the previous version of the manuscript. After considering their responses, I still believe that more can be done to address arm-level aneuploidies, or remoev it as it might be argued the authors are over-reaching. 

I understand and appreciate the authors have studied SMART-seq data to follow up, but this was not conclusive. I urge the authors to instead leverage 10x sequences, that they have already analyzed to call genetic variants from pooled reads and determine if candidate regions have show loss of heterozygosity compared to other region/candidate genes. This might be done in a similar fashion done to study GWAS arrays leveraging B-allele frequency to detect aneuploidies in immortalized cell lines.

---

## [Editor Report · Decision Letter 2]

22 Feb 2024

Dear Kim,

Thank you for the submission of your revised Short Report "Neuronal cell cycle re-entry events in the aging brain are more prevalent in neurodegeneration and lead to cellular senescence" for publication in PLOS Biology and thank you for addressing the last reviewer and editorial requests in this revision. On behalf of my colleagues and the Academic Editor, Jing Qu, I am pleased to say that we can in principle accept your manuscript for publication, provided you address any remaining formatting and reporting issues. These will be detailed in an email you should receive within 2-3 business days from our colleagues in the journal operations team; no action is required from you until then. Please note that we will not be able to formally accept your manuscript and schedule it for publication until you have completed any requested changes.

PRESS

Sincerely, 

Lucas Smith, Ph.D.

Senior Editor

PLOS Biology

lsmith@plos.org